# Nearshore fish community changes along the Toronto waterfront in accordance with management and restoration goals: Insights from two decades of monitoring

Sebastian Theis [1,2]*, Andrea Chreston [2], Angela Wallace [2], Brian Graham [2], Brynn Coey [2], Don Little [2], Lyndsay Cartwright [2], Mark Poesch [1], Rick Portiss [2], Jonathan Ruppert [2,3]

**1** Fisheries and Aquatic Conservation Lab, Faculty of Agricultural, Life and Environmental Sciences, University of Alberta, Edmonton, Alberta, Canada, **2** Toronto and Region Conservation Authority, Watershed Planning and Ecosystem Science, North York, Ontario, Canada, **3** Department of Ecology & Evolutionary Biology, University of Toronto, Toronto, Ontario, Canada

* theis@ualberta.ca

**Data Availability Statement:** Data and additional material are available through https://data.trca.ca/dataset/trca-waterfront-fish-data, https://trca-

## Abstract

Aquatic habitat in the Greater Toronto Area has been subject to anthropogenic stressors. The subsequent aquatic habitat degradation that followed led to the Toronto and Region waterfront being listed as an Area of Concern in 1987. Thus, extensive shoreline and riparian habitat restoration have been implemented as part of the Toronto and Region Remedial Action Plan in conjunction with local stakeholders, ministries, and NGOs in an overall effort to increase fish, bird, and wildlife habitat. A key aspect of current fish habitat restoration efforts, monitored by Toronto and Region Conservation Authority, is to account for long-term community changes within the target ecosystem to better understand overall changes at a larger spatial scale. Here we use electrofishing data from the past 20 years with over 100,000 records and across 72km of coastline to show how declines and fluctuations in fish biomass and catch along the waterfront are driven by a few individual species across three main ecotypes, such as coastal wetlands, embayments, and open coast sites, with the remaining species showing a high level of stability. Using community traits and composition for resident species we demonstrate native warmwater species have become more dominant along the waterfront in recent years, suggesting that restoration efforts are functioning as intended. Additionally, piscivore and specialist species have increased in their relative biomass contribution, approaching existing restoration targets. Altogether this waterfront-wide evaluation allows us to detect overall changes along the waterfront and can be beneficial to understand community changes at an ecosystem level when implementing and monitoring restoration projects.

camaps.opendata.arcgis.com, 10.6084/m9.figshare.24543517, and in the supplemental material. Contains information made available under the Toronto and Region Conservation Authority (TRCA)'s Open Data Licence v 1.0. Inquiries can be sent to opendata@trca.on.ca.

**Funding:** Funding for this project was provided by Mitacs Cluster Accelerate IT28524 to Dr. Mark Poesch. The funders had no role in study design, data collection and analysis, decision to publish, or preparation of the manuscript.

**Competing interests:** The authors have declared that no competing interests exist.

## 1. Introduction

Freshwater and its associated habitat that supports a wide diversity of biota is one of the most valuable resources in existence, providing numerous economic, social, and cultural services [1–4]. Impressively, while freshwater ecosystems only occupy 2.3% of the earth's surface, they support over 100,000 species (9.5% of total described animal species), sustaining many of the world's biodiversity hotspots [5–7]. Alarmingly, freshwater biodiversity declines regularly surpass those of terrestrial systems (83% between 1970 and 2014), which can be attributed to pollution, invasive species, habitat degradation, overharvest, climate change [2, 7]. Other lesser-known, emerging, threats include microplastics, e-commerce, algal blooms, and light and noise pollution [7]. Given the high value of freshwater ecosystems and the numerous threats they face, there has been a lot of work to address this issue through planning and development processes in many countries as well as on-the-ground restoration efforts [8–11]. Frameworks designed to mitigate negative impacts on ecosystems as well as to restore and enhance degraded freshwater systems require a thorough understanding of the current state of target ecosystems at a macroscale [9, 10, 12, 13].

Lake Ontario, having been impacted by the previously mentioned stressors, is a good example of the need of building a comprehensive current understanding of an ecosystem when considering past, current, and future restoration efforts [2, 11, 14]. For instance, Lake Ontario has been subject to many lake-wide events such as the introduction of non-native species like Round goby, Common carp, or dreissenid mussels [15–20]. These non-native species have led to ecosystem changes, ranging from out-competing native species to habitat degradation (e.g., Common carp [17, 21]). Other factors like increasing fluctuations in temperature or phyto- and zooplankton availability have a notable impact on specialist species like Alewife or Emerald shiner on which large parts of the food web rely as well as invertebrates like diporeia, or copepods [22–27].

The described lake-wide events and stressors can impact pelagic or littoral ecozones differently [28]. For example, power plants or runoff affect mainly nearshore waters, while non-native species' impact depends on their respective life history and habitat preferences for nearshore or offshore habitats [17, 20, 29]. Over time, Lake Ontario has experienced changes in nutrient cycling, with offshore habitat being deprived of nutrients while shifting higher nutrient loads toward the nearshore waters [26, 27, 30–32]. Changes in nutrient cycling combined with shoreline development, upwelling, and wind-induced turbidity, effectively divide nearshore from offshore waters. This division is also reflected in fish community composition, abundance, and health while changes in offshore and nearshore waters especially can be reflected in migratory or species with a large home range [26, 27, 31]. These complex and often difficult-to-detangle interactions and effects become essential when defining restoration targets and deciding on appropriate on-the-ground measures [11, 17, 33, 34]. A decline of specific species in one area, for example, nearshore Alewife occurrence, can be driven by impacts in another area like the lake-wide offshore decline in Alewife. Complex interactions might make restoration and enhancement efforts at the specific site less effective or vice versa mask the actual benefits of a restored site when focussing on species or metrics driven by outside factors [9, 11, 35–38].

The Toronto region has one of the highest population densities and is one of the largest urban areas in Canada (currently >50% urbanized within watersheds), representing an area with high pressure on remaining conservation areas and green spaces [15, 39–42]. Guided by the Toronto Waterfront Aquatic Habitat Restoration Strategy (TWAHRS), there have been approximately 55 ha of aquatic nearshore habitat restored on the Toronto Waterfront since the year 2000, with many more areas dating back decades [34, 43]. The Strategy as part of the

Toronto and Region Remedial Action Plan (RAP) outlines restoration techniques for various habitat types (open coast, estuary, embayment, and coastal wetland) and provides a framework for multi-agency collaboration and planning to improve habitat quality in those Areas of Concern (AOCs; [17, 24–26]). Barnes et al. (2020) assessed the overall effectiveness of the Strategy by examining habitat gains and changes in fish communities at each restored site [34]. While this comprehensive work provided an assessment of changes in fish populations related to the specific site-level objectives, it did not fully examine the overall changes in the nearshore fish community along the Toronto Waterfront as a whole. While each restoration site along the Toronto Waterfront has site-specific objectives and goals, their combined cumulative effect is equally important to be evaluated and assessed. Community assessments on a large temporal and spatial scale can be an invaluable indicator for overall community and ecosystem health and stability and help put documented restoration effects in the broader nearshore or even lake-wide context [13, 28, 44, 45].

Our objective with the broad spatiotemporal scale monitoring data from the Toronto Waterfront nearshore community is to determine how the fish populations along the Toronto Waterfront have changed over the past two decades, expanding on previous timeframes (previously up to 2018) under consideration of:

- Catch and biomass,

- Most abundant and biomass-contributing species,

- Community life history and habitat traits,

- Dominant ecotypes (open coast, coastal wetland, embayment).

Specifically, we aim to determine if said community changes align with the most prominent stated restoration goals [33, 46] to:

1. **Population trends:** Increase stability for catch and biomass trends along the waterfront.

2. **Community changes:** Increase specific community trait presence: native piscivore and specialist species presence (reduce non-native species presence), foster native, warmwater, and vegetation-associated communities.

3. **Specialist and piscivore targets:** Reach 20% of total biomass attributed to piscivores and 40% of total biomass attributed to specialist species.

Here this work will address multiple gaps in knowledge by conducting a broad spatiotemporal analysis of community trends and changes based on data synthesis, analyzing key relationships of species and associated traits that are related to biodiversity and fish community changes. This information is beneficial to both managers and decision-makers to better inform and evaluate if the nearshore community along the Toronto Waterfront is approaching the outlined restoration targets and desired community composition.

## 2. Methods

### 2.1 Data

Toronto and Region Conservation Authority (TRCA) is responsible for environmental protection and restoration in within the region with 72km of shoreline falling into their jurisdiction (Fig 1).

Fish community data collected during TRCA surveys is stored in a centralized database. Fish population trends and community analyses for the Toronto waterfront between the years of 2003 to 2021 are based on 1,354 boat electrofishing runs conducted by TRCA using either a

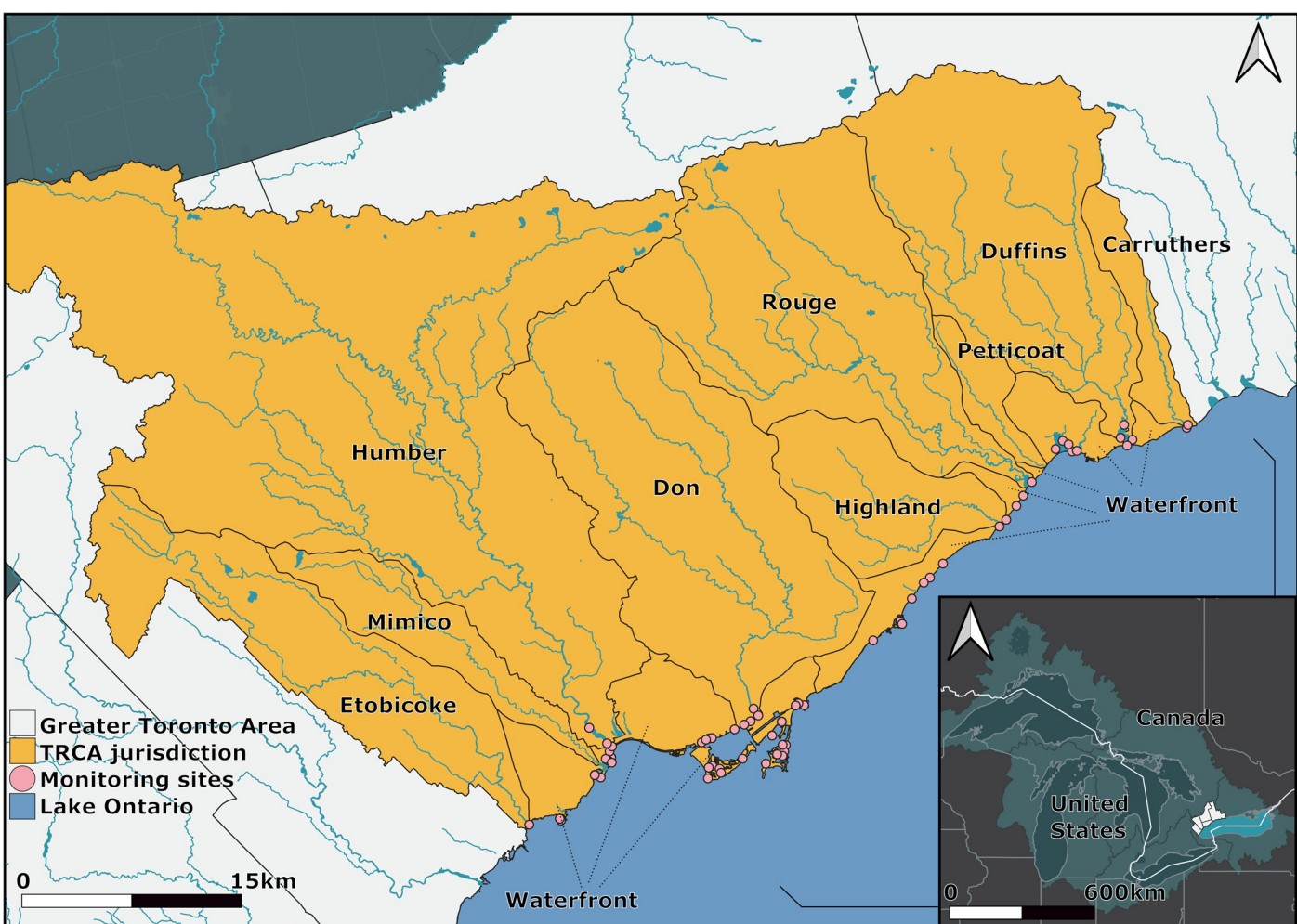

**Fig 1. TRCA jurisdiction.** Toronto and Region Conservation Authority jurisdiction spans 72km along the Toronto waterfront, encompassing 44s of near shore restoration projects and 238 sites (spatial data source: TRCA/ https://data.trca.ca/; data to be used under the Toronto and Region Conservation Authority (TRCA)'s Open Data Licence v 1.0.; base layers available in http://www.qgis.org/; Map created using the Free and Open Source QGIS).

14' or 18' (21' instead of 18' fall 2021 onward) Smith-Root electrofishing boat. Only runs in embayments, open coast, and coastal wetland sites were considered as these are the three predominant habitat types along the waterfront, and to keep analyses comparable. Summer data collected between 2003 and 2021 were used due to the highest data availability and standardization for these years and seasons. Most monitored sites are restored and reflect the overarching effort to restore the waterfront through a network of 44 restoration projects (238 sites) along the 72km of shoreline falling within TRCA jurisdiction (Fig 1). There were 192 summer runs in coastal wetlands (11,986 individual fish), 691 in embayments (28,525 individual fish), and 471 for open coast sites (12,540 individual fish) along the waterfront (S1 Fig for waterfront fish species and scientific names).

Fish data were available as catch in each electrofishing run, identified to the species level, length (mm), and weight (g). Collected fish data per run allowed us to calculate catch per unit effort (CPUE; standardized to 1000 seconds of shock time per run), and biomass per unit effort (BPUE; standardized to 1000 seconds of shock time per run).

## 2.2 Statistical analyses

**2.2.1 Population trends over time.** Population trends for fish communities were modeled in two ways, first by using species abundance as estimated through electrofishing catch within a run (n/1000s; CPUE) and second by using species biomass as estimated through fish weight per run (g/1000s; BPUE). These population metrics were used to identify changes over time along the waterfront between 2003 and 2021 in the summer and across the three ecotypes (open coast, embayments, and coastal wetlands). Years were divided into three periods of time and labeled as past (03–08; T1), intermediate (09–14; T2), and current (15–21; T3) communities. Choosing broad time increments allows us to characterize past and present communities while acknowledging a long-term and ongoing restoration process as well as potential biases within years (e.g., high water, closed sections, phragmites management, sampling errors, etc.). These, often contribute to stochastic responses in metrics that can skew sampling within a given year, but overall community changes will still be captured within the context of multi-year intervals ([11, 47]; S2 Fig).

Median CPUE and BPUE (first quartile Q1, third quartile Q3) were compared across the three time periods for each ecotype through Kruskal-Wallis tests (Alpha < 0.05) and pairwise comparisons for the three time periods (Conover-Iman if $H_0$ rejected, Holm-Bonferroni adjusted p-values; conover.test; [48]). The median was chosen based on the variability of CPUE and BPUE across runs to ensure central tendencies for these skewed distributions [49]. Species contributing the most to overall CPUE or BPUE were determined (% contribution to overall CPUE and BPUE per run) and the same analysis was run without them to discern if overall community and population trends were potentially driven by a few individual species and if these species are related to restoration efforts, near-shore community, and lake-wide events.

**2.2.2 Community traits and habitat associations along the Toronto waterfront–RDA.** Community changes for this project were based on relative abundance per species (Hellinger-transformation). Relative abundance can be a helpful metric when trying to capture community changes. As opposed to CPUE, which is meant to estimate species abundance, relative abundance measures the proportion of each species within a sample run and across ecotypes [50, 51]. We could then assess what the fish community looks like in an average electrofishing run at an ecotype and year. This can be helpful since increases or decreases in catch (for each sampling run) can mask proportionate changes and shifts in communities, especially with high catch differences among runs and with highly abundant species that can inflate overall catch (e.g., Alewife; [14]; S3 Fig). For instance, a specific species or type can have the same abundance across two sites but very different relative abundance depending on the rest of the site community.

Relative abundance (Hellinger transformed data) was plotted in ordination space (Redundancy Analysis (RDA)) and compared across ecotypes and the three main time intervals. The utility of RDA is that it simplifies complex data and relations, while keeping trends/patterns and works best for strongly correlated variables [52], which is the case for community characteristics, as defined by species traits and time (years; vegan; [53]). Dominant species and associated traits characterizing the current community at a specific ecotype and time allow us to check if said community represents the one aimed for in the official restoration goals.

**2.2.3 Specific biomass targets for piscivores and specialists.** Specific biomass targets were analyzed by median biomass proportion of total biomass for a given year and compared through Kruskal-Wallis tests (Alpha < 0.05; Dunn post-hoc if $H_0$ rejected, BH adjusted p-values) across the three main periods (conover.test; [48]). Official targets were a 20% biomass contribution to total biomass by piscivore species and 40% by specialist species along the

waterfront (S4 Fig). Piscivore species in the context of the Toronto and Region RAP were top-piscivores, predominantly feeding on other fish whereas specialist species cover trophic specialists, mostly relying on a single feeding guild that is not piscivore [33, 34].

# 3. Results

## 3.1 Population trends

From here on periods will be referred to as followed: T1 (2003 to 2008), T2 (2009 to 2014), T3 (2015 to 2021). For detailed statistics and results tables please refer to the supplemental section of this paper (S1–S5 Tables).

**3.1.1 CPUE trends over time and ecotypes.** CPUE for all three ecotypes changed over time comparing T1 and T3 as well as T2 and T3. Notably, there were no significant changes recorded between T1 and T2 (Fig 2).

CPUE for coastal wetlands was driven by Alewife and Emerald shiner, with initial significant decreases in median CPUE (chi-squared = 7.409, df = 2, p-value = 0.02) disappearing after removing Alewife and Emerald shiner (chi-squared = 2.1248, df = 2, p-value = 0.35; Fig 2A). Alewife and Emerald shiner median CPUE declined over time (chi-squared = 6.334, df = 2, p-value = 0.04), especially between T2 and T3 ($p < 0.05$; Fig 2A), decreasing by more than 50%. Compared to the other two ecotypes, coastal wetlands had the highest median CPUE ($\sim$20–40% higher on a run basis), ranging between 57.6 and 77.4/1000s.

CPUE at embayments differed across time periods (chi-squared = 7.2177, df = 2, p-value = 0.03), decreasing from a median of 62.5/1000s (T1) to 53/1000s (T1-T2 p = n.s.) and 31.3/1000s (T2-T3 & T1-T3 $p < 0.05$; Fig 2B). Like open coast sites, CPUE was driven by Alewife and Emerald shiner, both declining significantly over time (chi-squared = 6.6511, df = 2, p-value = 0.04). Consequently, removing those two species from the CPUE analyses left CPUE for the remaining species at embayments stable over time (chi-squared = 3.0803, df = 2, p-value = 0.21; Fig 2B).

Open coast median catch across the three time periods stayed stable (chi-squared = 4.5902, df = 2, p-value = n.s.), ranging from 20.5/1000s (T1) to 34/1000s (T2) and 21/1000s (T3; Fig 2C). CPUE at open coast sites was largely driven by Alewife and Emerald shiner. Both species decreased significantly in their CPUE at open coast sites (chi-squared = 5.8256, df = 2, p-value = 0.04), specifically between T2 and T3 ($p < 0.05$; Fig 2C).

**3.1.2 BPUE trends over time and ecotypes.** Coastal wetlands, having the highest median BPUE of all three ecotypes ($\sim$5–35% higher on a run basis), did not decrease significantly in their median BPUE over the three time periods with BPUE declining slightly from 14179g/1000s (T1) to 12676g/1000s (T2) and finally 10514g/1000s (T3; chi-squared = 4.8391, df = 2, p-value = n.s.; Fig 3A). The highest contributing species in terms of biomass were Common carp and Freshwater drum. Both species declined in BPUE over time (chi-squared = 6.5992, df = 2, p-value = 0.04), comparing T1 and T3 and T2 and T3 ($p < 0.05$; Fig 3A).

Sheltered embayments along the Toronto waterfront decreased in median BPUE over time (chi-squared = 7.197, df = 2, p-value = 0.03; Fig 3B) from 13632g/1000s (T1) to 8501g/1000s (T1-T2; p = n.s.) and 6896 (T1-T3, $p < 0.05$, T2-T3 p = n.s.). Removing the highest contributing species (Common carp and White sucker), showed no significant biomass changes for embayment over time (chi-squared = 2.2301, df = 2, p-value = 0.33), whereas Common carp and White sucker biomass declined over time (T1-T2 p < n.s., T1-T2 & T2-T3 $p < 0.001$; Fig 3B).

Biomass differences were significant across the three time periods for open coast sites (chi-squared = 9.4233, df = 2, p-value = 0.01; Fig 3C). Open coast median BPUE increased from 3459g/1000s (T1) to 4801g/1000s (T1-T2 $p < 0.05$) and decreased to 1616g/1000s in period T3 (T2-T3 $p < 0.05$; T1-T3 $p < 0.05$). The highest contributing species for open coast site biomass

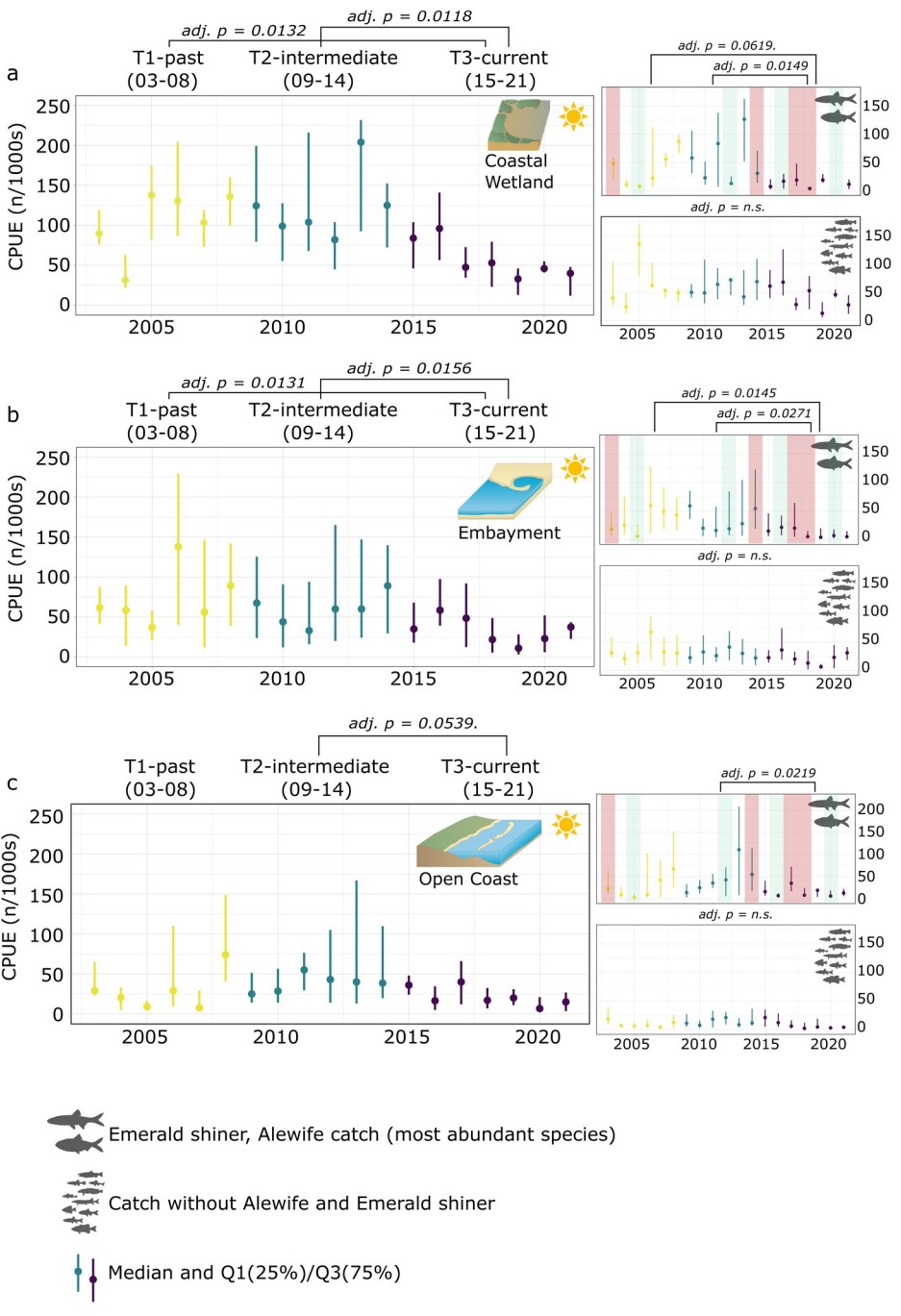

**Fig 2. Population trends for catch.** Median catch (Q1, Q3, CPUE (n/1000s) across time (2003–2021) along the Toronto waterfront and across the three ecotypes (a coastal wetland, b embayment, c open coast). Kruskal-Wallis test and conover comparison (BH adjusted p-value) for years blocked into three periods (T1 03–08, T2 09–14, T3 15–21). Comparisons were done for overall catch as well as excluding species contributing the most to overall catch for each ecotype (Emerald shiner, Alewife for all three ecotypes). As the most abundant species, strong (green) and weak (red) reproductive Alewife years were highlighted based on pelagic trawling surveys [25]. Symbol attribution Tracey Saxby, Integration and Application Network; Kate Moore, Moreton Bay Waterways and Catchments Partnership (ian.umces.edu/media-library). Reprinted from ian.umces.edu/media-library under a CC BY 4.0 license, with permission from ian.umces.edu/media-library, original copyright 2005 & 2010.

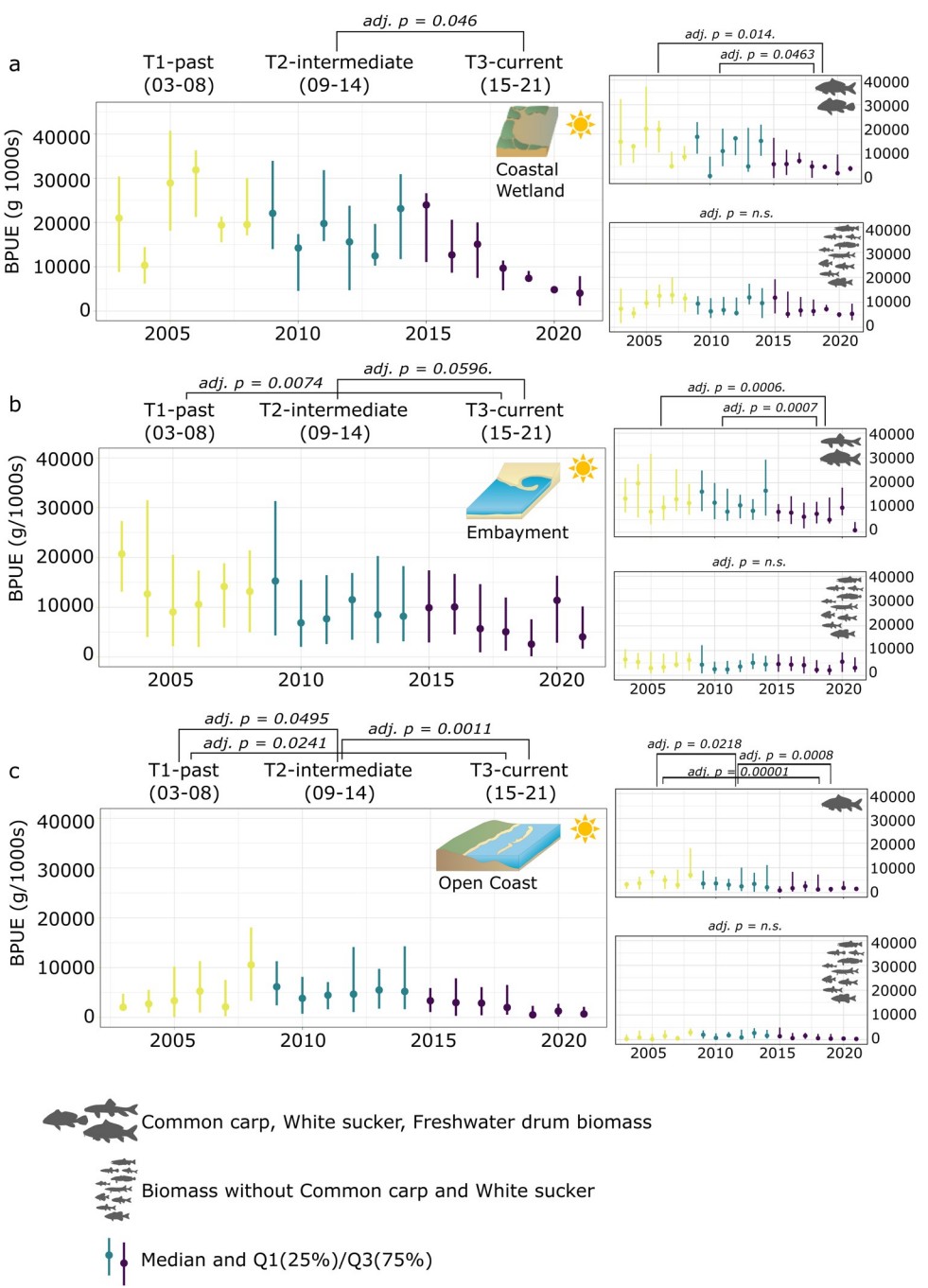

**Fig 3. Population trends for biomass.** Median biomass (Q1, Q3, BPUE, g/1000s) across time (2003–2021) along the Toronto waterfront and across the three ecotypes (a coastal wetland, b embayment, c open coast). Kruskal-Wallis test and conover comparison (BH adjusted p-value) for years blocked into three periods (T1 03–08, T2 09–14, T3 15–21). Comparisons were done for overall catch as well as excluding species contributing the most to overall biomass for each ecotype (Common carp (open coast, coastal wetland, embayment), White sucker (embayment), and Freshwater drum (coastal wetland). Symbol attribution Tracey Saxby, Integration and Application Network; Kate Moore, Moreton Bay Waterways and Catchments Partnership (ian.umces.edu/media-library). Reprinted from ian.umces.edu/media-library under a CC BY 4.0 license, with permission from ian.umces.edu/media-library, original copyright 2005 & 2010.

was Common carp. Removal of carp biomass from open coast sites showed no significant biomass changes over time (chi-squared = 3.7391, df = 2, p-value = 0.15), while Common carp biomass at open coast sites declined steadily over the three time periods, being the only ecotype with notable biomass changes between T1 and T2 (T1-T2 p < 0.05, T1-T3 & T2-T3 p < 0.001; Fig 3C).

## 3.2 Community changes

**3.2.1 Community trait and composition changes based on abundance.**  Coastal wetlands showed the most pronounced community trait changes based on relative species abundance across all three ecotypes (RDA1 40.15%, RDA2 26.06%, df = 2, F-value = 3.875, p < 0.05, Fig 4A). All three time periods differed from each other (p < 0.05), with a distinct change towards a higher abundance of vegetation and cover-associated species, often native warmwater species as well as an increased dominance of piscivores (T2 & T3, e.g., Largemouth bass, Golden shiner, Pumpkinseed, Bluegill).

Embayment communities changed over time with all three time periods being different from each other (RDA1 40.81%, RDA2 22.45%, df = 2, F-value = 4.121, p < 0.05; Fig 4B). Earlier dominance of open water cool- and coldwater specialists like Alewife and Emerald shiner (T1) were replaced by an increase in cool, warmwater, cover, and vegetation-associated species and native generalists and piscivores like Largemouth bass, Brown bullhead, Gizzard shad and in some years non-native Round goby.

Open coast community traits and composition changed over time based on the RDA results across the three time periods (RDA1 36.36%, RDA2 23.44%, df = 2, F-value = 2.069, p < 0.05), with significant differences between T1 and T3 (p < 0.05, Fig 4C). Open coast communities changed towards a higher dominance of native species and piscivore species (e.g., Smallmouth bass, Rock bass) with cool and coldwater preferences in T2 and T3 compared to T1.

**3.2.2 Community trait and composition changes based on biomass.**  Coastal wetland relative biomass and associated community traits differed over the three time periods (RDA1 35.69%, RDA2 22.25%, df = 2, F-value = 3.559, p < 0.05, Fig 5A). The community shifted from non-native and generalist dominance (e.g., Common carp, Freshwater drum, and Alewife (coldwater)) towards higher relative biomass of native piscivores like Largemouth bass and Bowfin (T3). The relative biomass of warmwater, coolwater and vegetation, and cover-associated species also increased (T2 & T3; Fig 5A).

Embayment community traits changed over time, with the largest differences observed between T2 and T3 as well as T1 and T3 based on relative biomass (RDA1 50.62%, RDA2 19.24%, df = 2, F-value = 4.234, p < 0.05, Fig 5B). While earlier periods were associated with a high relative non-native generalist biomass like Common carp or coldwater species like Alewife, T3 showed a shift towards native specialists and piscivores like Yellow perch, Bowfin, and a larger relative biomass of Gizzard shad (Fig 5B).

Open coast community traits based on relative biomass changed little over the three time periods (RDA1 40.51%, RDA2 15.37%, df = 2, F-value = 1.473, p = 0.16, Fig 5C). Smaller changes over time occurred in open coast communities towards higher cold and coolwater biomass as well as native piscivores and specialist species through a reduction in Common carp relative biomass and an increase in relative biomass for various salmonids (Fig 5C).

## 3.3 Specialist and piscivore targets

**3.3.1 Piscivore proportionate biomass contribution across ecotypes.**  The contribution of piscivores to total biomass in coastal wetlands increased over the three time periods from 14% in T1 to 21% in T3, exceeding the 20% mark (Fig 6A). Species richness for piscivores

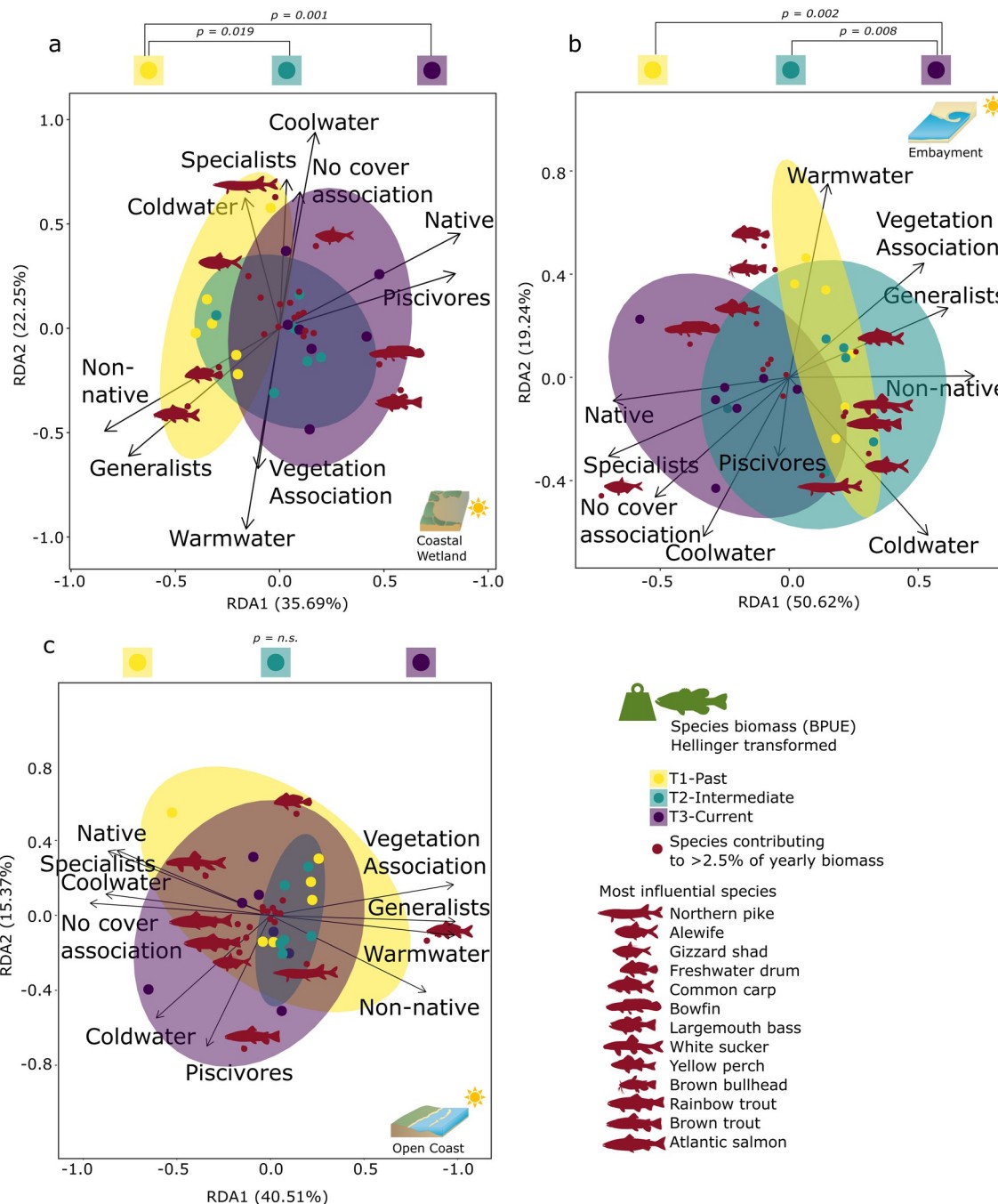

**Fig 4. Community changes based on catch.** Community traits and changes along the Toronto waterfront in ordination space (RDA) across the three ecotypes (a coastal wetland, b embayment, c open coast) based on species CPUE (Hellinger transformed). Years are blocked into three periods through ellipses (T1 03–08, T2 09–14, T3 15–21) and compared through adonis analysis. Highlighted species cover the most influential species, contributing to community changes (species contributing less than < 2.5% to yearly catch were removed). Symbol attribution Tracey Saxby, Integration and Application Network; Kate Moore, Moreton Bay Waterways and Catchments Partnership (ian.umces.edu/media-library). Reprinted from ian.umces.edu/media-library under a CC BY 4.0 license, with permission from ian.umces.edu/media-library, original copyright 2005 & 2010.

doubled from an initial 5 species in T1 to 10 species in T3, with species such as American eel (T2, T3) or Burbot (T3) being recorded over time.

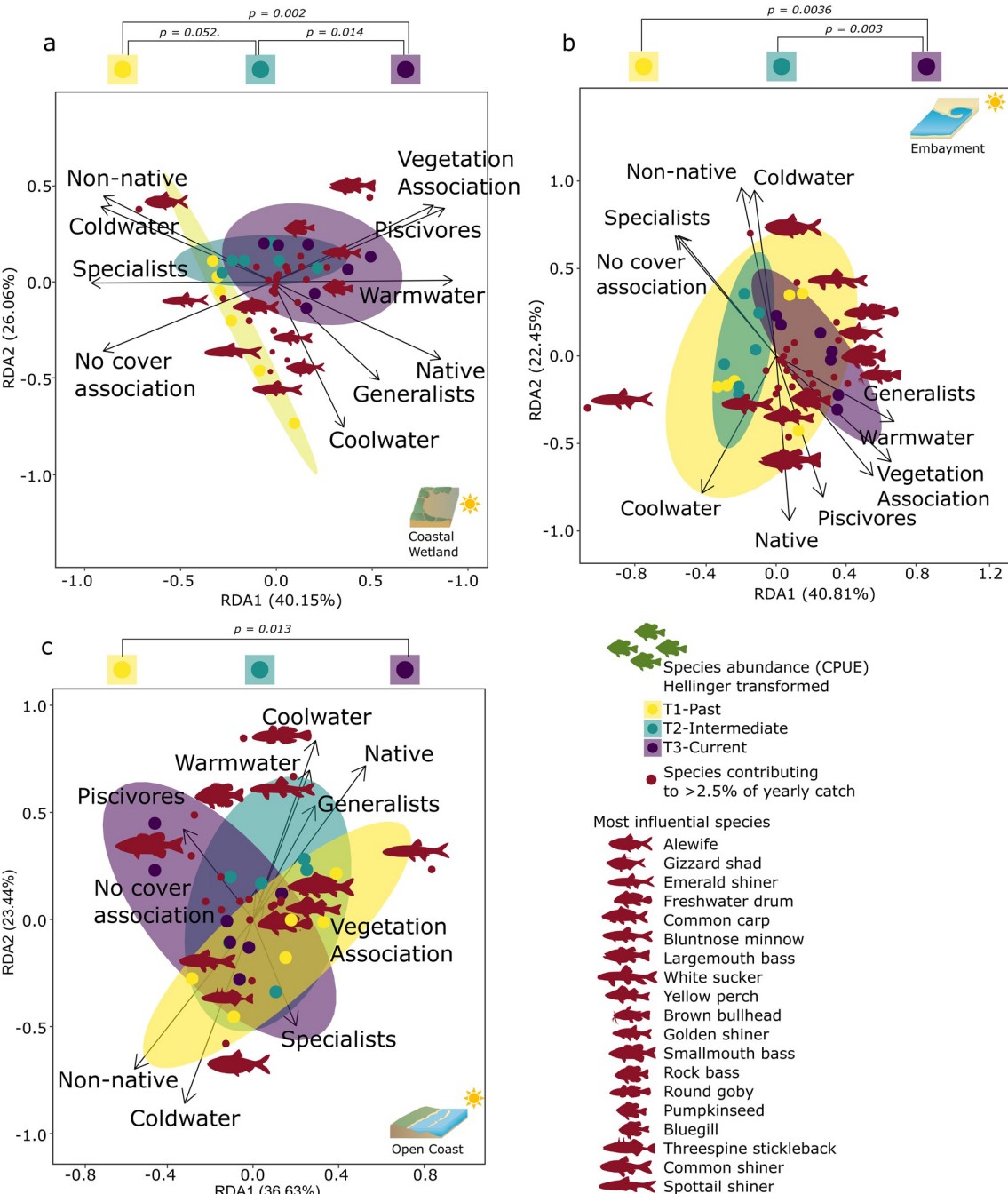

**Fig 5. Community changes based on biomass.** Community traits and changes along the Toronto waterfront in ordination space (RDA) across the three ecotypes (a coastal wetland, b embayment, c open coast) based on BPUE (Hellinger transformed). Years are blocked into three periods through ellipses (T1 03–08, T2 09–14, T3 15–21) and compared through adonis analysis. Highlighted species cover the most influential species, contributing to community changes (species contributing less than < 2.5% to yearly biomass were removed). Symbol attribution Tracey Saxby, Integration and Application Network; Kate Moore, Moreton Bay Waterways and Catchments Partnership (ian.umces.edu/media-library). Reprinted from ian.umces.edu/media-library under a CC BY 4.0 license, with permission from ian.umces.edu/media-library, original copyright 2005 & 2010.

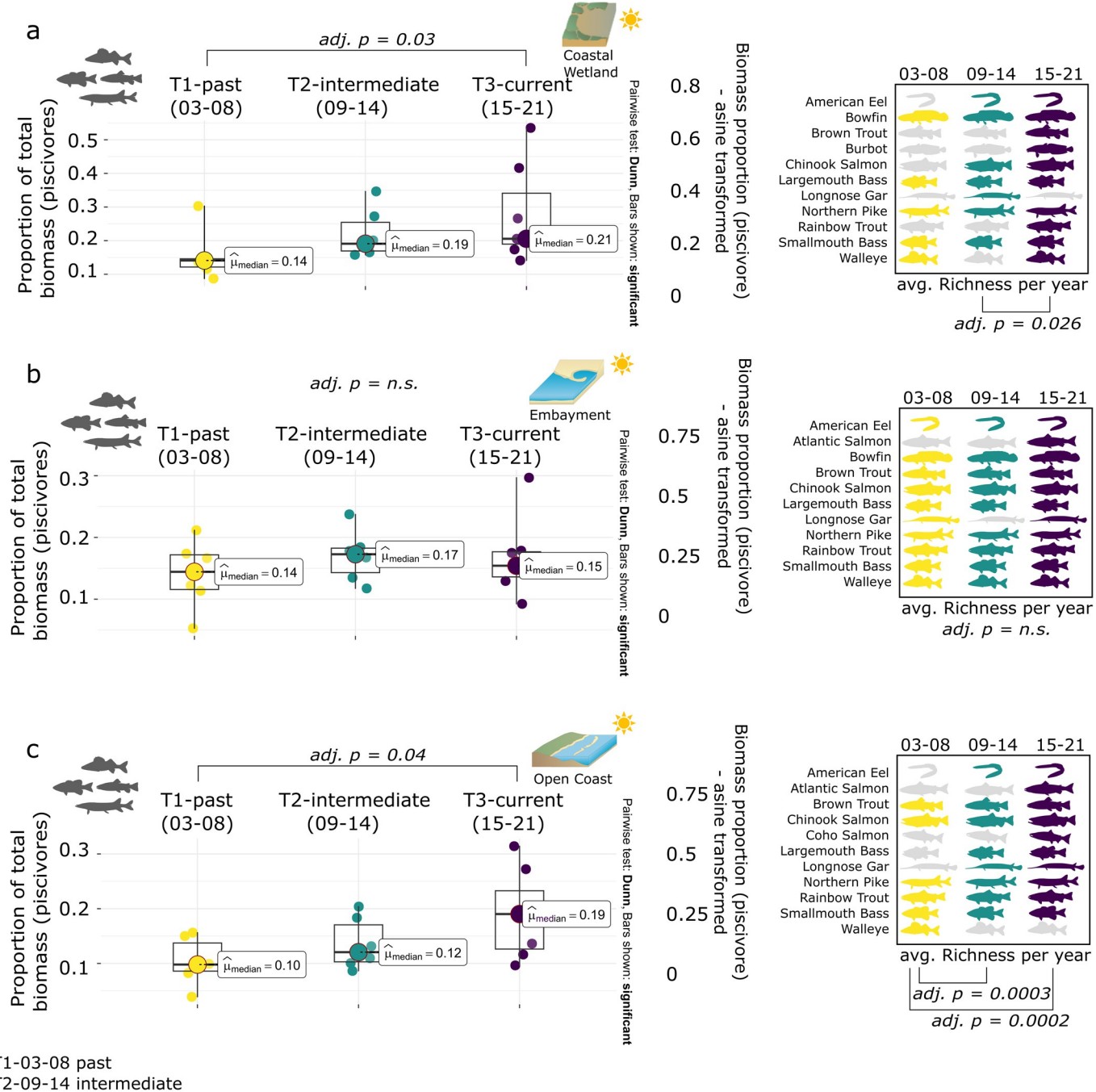

**Fig 6. Piscivore biomass targets.** Median piscivore biomass contribution to yearly total biomass (Q1, Q3) along the Toronto waterfront across the three ecotypes (a coastal wetland, b embayment, c open coast) as well as average yearly piscivore richness. Kruskal-Wallis test and conover comparison (BH adjusted p-value) for years blocked into three periods (T1 03–08, T2 09–14, T3 15–21). Symbol attribution Tracey Saxby, Integration and Application Network; Kate Moore, Moreton Bay Waterways and Catchments Partnership (ian.umces.edu/media-library). Reprinted from ian.umces.edu/media-library under a CC BY 4.0 license, with permission from ian.umces.edu/media-library, original copyright 2005 & 2010.

Median piscivore biomass contribution for embayment sites was stable over time ranging between 14 and 17%. Piscivore richness also stayed consistent with 9 to 11 different species between T1 and T3 (Fig 6B).

Median piscivore biomass contribution for open coast sites increased over time, increasing from 10% (T1) to 12% (T2) and 19% (T3), approaching the 20% benchmark (Fig 6C). Increases in biomass contribution by piscivore species at open coast sites coincided with an overall increase in piscivore richness (T1 n = 6, T2 n = 8, T3 n = 10, e.g., Largemouth bass, American eel, Coho salmon).

**3.3.2 Specialist proportionate to biomass contribution across ecotypes.**  Coastal wetland sites had the lowest specialist biomass proportion and overall specialist richness of the three ecotypes, slowly increasing from 19% in T1 to 25% in T3, distributed across a total of 27 different species (Fig 7A). Within specialist biomass changes over time were driven by an increase in herbivores (+9% median biomass contribution; Gizzard shad, Fig 8).

Embayments had the highest richness of specialist species (n = 34) which stayed consistent over time, with biomass contribution to total biomass ranging around the 40% target (37% T1, 32% T2) while meeting it in T3 (40%, Fig 7B). Specialist biomass changes at embayments were attributed to an increase in herbivores (+17% median biomass contribution; Gizzard shad) and a decrease in invertivores (-11% median biomass contribution; White sucker, Fig 8).

Median specialist biomass contribution for open coast sites was high for all three time periods, exceeding the 40% mark, increasing slightly from 41% in T1 to 52% in T3 while recording a total of 31 different species during those periods (no increase over time, Fig 7C). Specialist biomass changes at open coast sites were mainly due to an increase in herbivores (+5% median biomass contribution; Gizzard shad) and a decrease in planktivores (-5% median biomass contribution; Alewife) and generalists (-14% median biomass contribution; Common carp, Fig 8).

## 4. Discussion

Our main findings can be summarized regarding the three main objectives as followed:

- Fish population trends over the past 20 years along the Toronto waterfront are mainly driven by a few individual species (Alewife and Emerald shiner for abundance, Carp for biomass), while the rest of the populations show more stability in terms of CPUE and BPUE.

- Fish communities along the waterfront based on relative catch and biomass have changed towards an increased dominance of native warmwater species, preferring vegetation and cover as well as an increase in piscivore presence. Open coast sites, compared to coastal wetlands and embayments show a higher presence of cool and coldwater species.

- Median piscivore biomass contribution is approaching or meeting the 20% mark for coastal wetlands and open coast sites coinciding with an increase in overall and average piscivore species richness while being consistently around 15% for embayment sites with stable high overall and average richness.

- Median specialist biomass contribution has not changed significantly for any of the three ecotypes but is high for embayment and open coast sites, meeting the 40% target, while being lower (25%) at coastal wetlands.

### 4.1 Do population trends align with restoration goals and lake-wide events?

Population trends captured by the summer boat electrofishing data for the three ecotypes along the Toronto waterfront show that fish populations were rather stable over the past two decades,

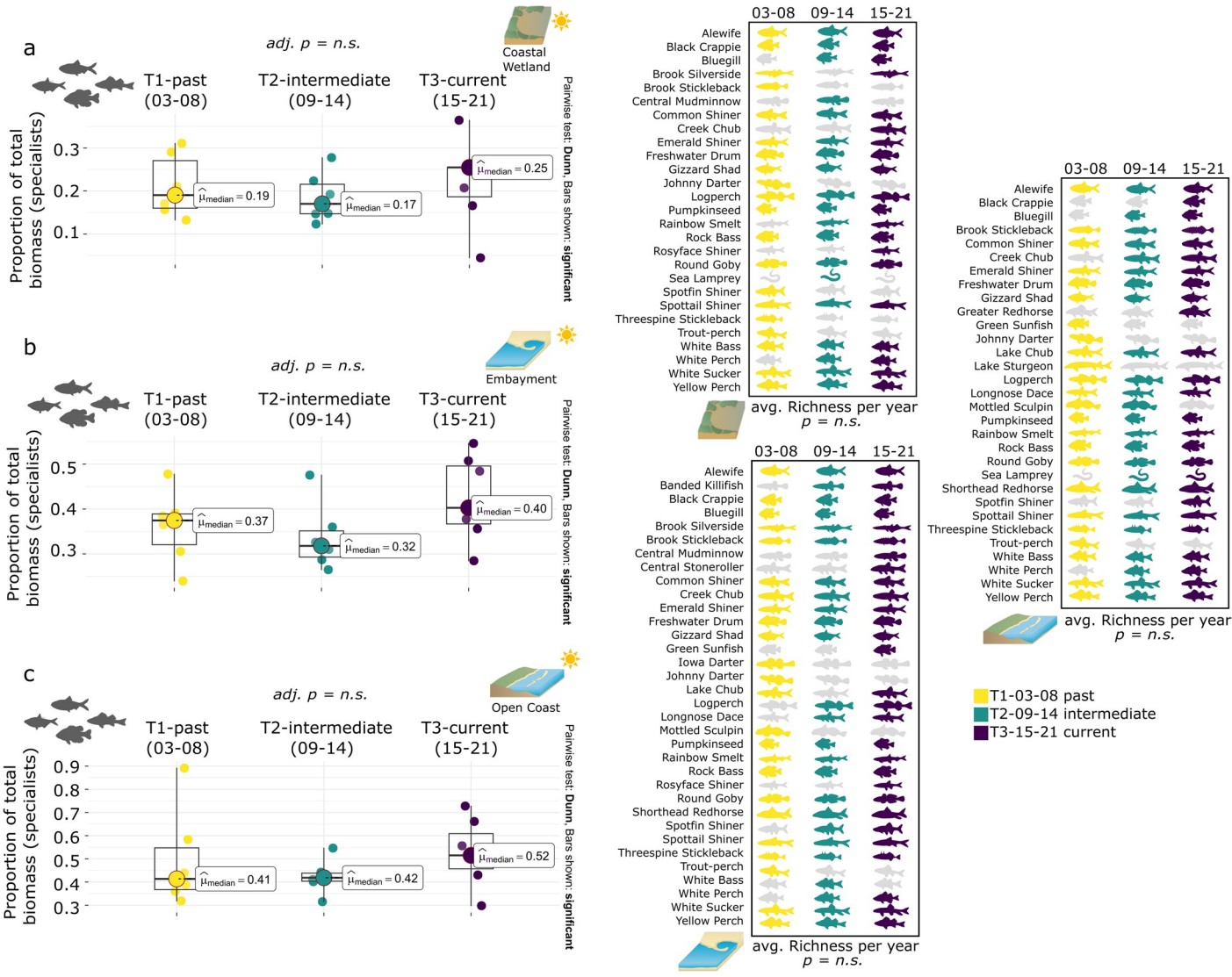

**Fig 7. Specialist biomass targets.** Median specialist biomass contribution to yearly total biomass (Q1, Q3) along the Toronto waterfront across the three ecotypes (a coastal wetland, b embayment, c open coast) as well as average yearly specialist richness. Kruskal-Wallis test and conover comparison (BH adjusted p-value) for years blocked into three periods (T1 03–08, T2 09–14, T3 15–21). Symbol attribution Tracey Saxby, Integration and Application Network; Kate Moore, Moreton Bay Waterways and Catchments Partnership (ian.umces.edu/media-library). Reprinted from ian.umces.edu/media-library under a CC BY 4.0 license, with permission from ian.umces.edu/media-library, original copyright 2005 & 2010.

both in terms of BPUE and CPUE when accounting for highly variable species like Alewife, aside from naturally occurring fluctuations and probable sampling bias [24, 28, 54, 55].

Overall fluctuations and declines in certain areas and species are well documented and align with different factors affecting the Lake Ontario nearshore community like notable invasions of Round goby [56], dreissenid mussel [28], or the continuous impact of Common carp [18]. Wide variation in Alewife cohorts in our nearshore data is supported by an overall decline in populations as confirmed by pelagic trawling data, associated with temperature, zooplankton abundance and predation dynamics [23–25]. Abundance declines are also due to an overall shift in the food web over time and changes in nutrient input as well as anthropogenic development [15, 16, 21, 28, 57].

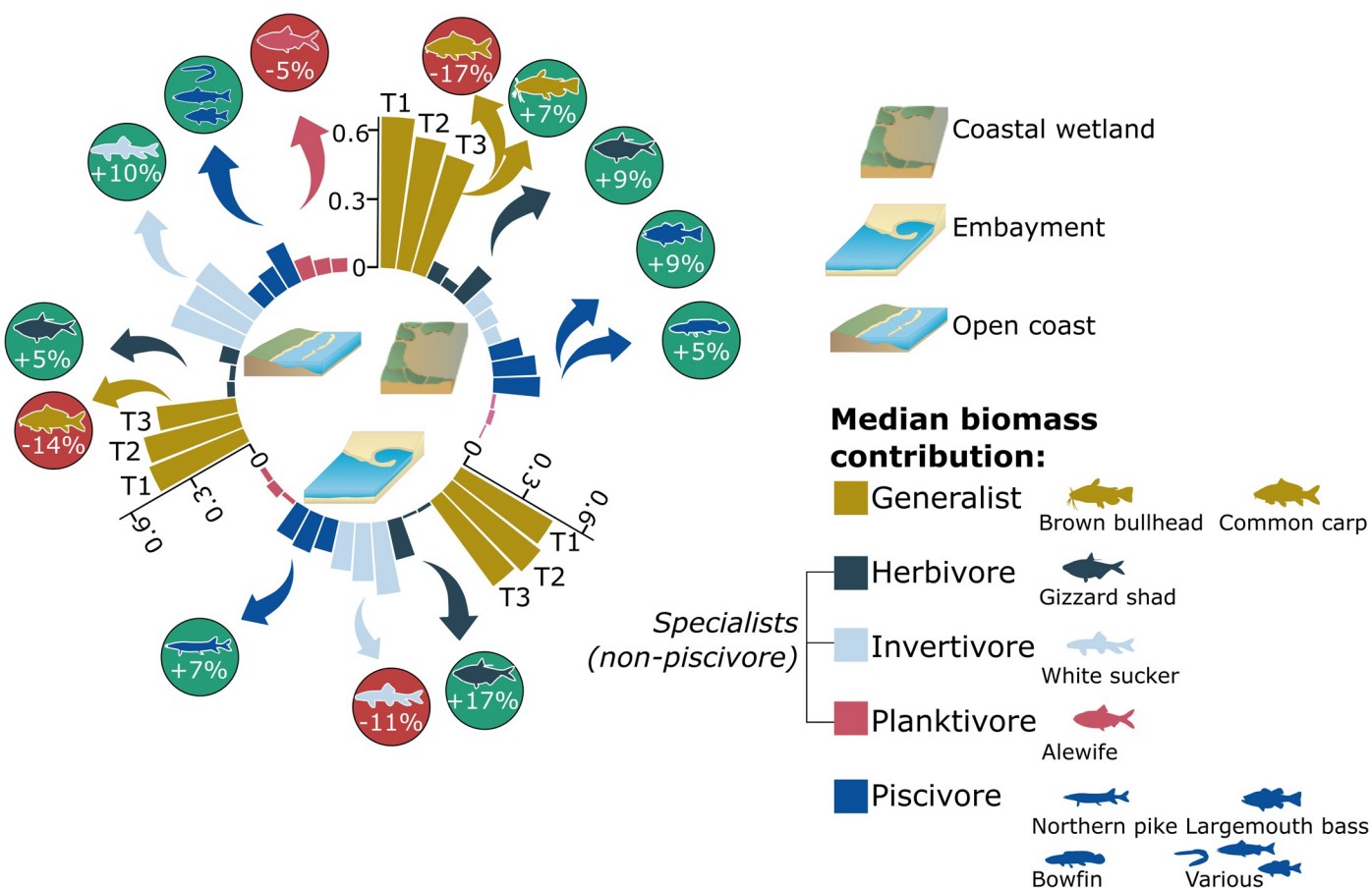

**Fig 8. Detailed biomass contribution by trophic guild.** Median biomass contribution to yearly total biomass by primary trophic guild (and influential species) along the Toronto waterfront across the three ecotypes (coastal wetland, embayment, open coast). Years are blocked into three periods (T1 03–08, T2 09–14, T3 15–21). Symbol attribution Tracey Saxby, Integration and Application Network; Kate Moore, Moreton Bay Waterways and Catchments Partnership (ian.umces.edu/media-library). Reprinted from ian.umces.edu/media-library under a CC BY 4.0 license, with permission from ian.umces.edu/media-library, original copyright 2005 & 2010.

Declines in Emerald shiner, aside from their wide yearly natural fluctuations, are also common in the presence of strong Alewife and Gizzard shad cohorts given competition for zooplankton and lower temperature thresholds compared to other more tolerant species [22, 58]. While Alewife have been declining along the waterfront, Gizzard shad have been steadily increasing at many sites. Furthermore, Emerald shiner prefer cool, open waters in a lentic environment which is the opposite of the habitat present at many of the restored sites and ecotypes [59]. Declines in White sucker abundance are directly linked to competition with Round goby which is well documented for Lake Ontario [16, 60, 61].

Common carp, as the main biomass contributor at all three ecotypes, has been declining along the waterfront which is an encouraging finding. Overall, declines can be mainly attributed to large-scale physical efforts to keep Common carp out of certain areas like coastal wetlands (exclusion barriers) as well as the beneficial effect of restored sites on the overall stability of the rest of the fish community [21, 33, 62]. Reduction in Common carp biomass and abundance however needs to be considered with a caveat related to physical exclusion barriers used for some of the monitored coastal wetlands. Barriers can prevent fish of certain sizes to enter a wetland but only excludes said fish from the specific area but not the rest of the ecosystem [21]. Many monitored sites along the waterfront are restored so it is necessary to observe

Common carp trends, as well as other non-native species within the rest of the nearshore community to infer ecosystem-wide trends [14, 33, 63]. Telemetry studies are one effective way to follow the movement and behavior patterns of non-native species [18].

Restored areas along the waterfront seem to benefit the overall fish communities across different ecotypes in terms of biomass and abundance stability, thus supporting the maintenance of healthy, diverse fish communities. Declines and fluctuations are driven by a few individual species for which most are not restoration related (e.g., Alewife) and others like Common carp, where observed declines are a positive outcome.

## 4.2. Do community changes align with restoration goals?

Abundance and biomass decline of Common carp, Alewife, Emerald shiner, White sucker, and Freshwater drum, shown in the previous population trends, are also reflected in community trait changes based on relative CPUE and BPUE. Community changes based on assessed community traits and community composition show that coastal wetland and embayment sites have changed significantly over time along the waterfront. Both ecotypes have a stronger association with native warmwater and vegetation-loving species as well as piscivore species in recent years. Embayments field a broader mix of warmwater and coolwater species, which aligns with the overall role of embayments of providing shelter for a mix of cool and warmwater species. Species diversity and a more frequent turnover in embayments is due to their general physical habitat characteristics of vegetated and cover-rich riparian areas and more open deeper coolwater areas as well as being exposed to upwellings [33, 64, 65].

These changes are encouraging to see, especially given the increased dominance of native species in both ecotypes like Bluegill, Pumpkinseed, Yellow perch, Bowfin, or Rock bass. These community changes are to be expected since many of the restored sites employ restoration methods like placement of coarse woody habitat, boulders as well as riparian planting. These methods are aimed at providing habitat for a mixed community of warm and coolwater species, with a preference for cover and vegetation as well as providing forage fish supplies for piscivore species (e.g., Largemouth bass, Northern pike; [33, 59, 65]).

Open coast sites stayed very consistent in their community trait distribution, with slight biomass and abundance changes towards cool and coldwater species as well as native piscivores. Natural open coast habitat, less productive as seen in the catch data compared to coastal wetlands and embayments, is meant to support a different type of fish community, aimed at cool and coldwater species, hence meeting its desired goals, with increased dominance of Smallmouth bass and salmonid species [33, 59, 66]. Stability for open coast sites also speaks to the fact that many of these sites have been restored for a longer period, showing the effectiveness of the applied restoration methods such as surcharged open coast revetments [33, 67].

**4.2.1 Piscivore and specialist biomass.** The cumulative effect of population trends and community composition changes are reflected in the total biomass contribution of piscivore and specialist species, an important benchmark for evaluating trophic structure and balance [46, 68–70]. Overall, all three ecotypes support a balanced trophic structure with median piscivore biomass making up 15 to 21% of total biomass in recent years and an increase in piscivore richness. A balanced trophic structure will help to ensure overall ecosystem stability and improve ecosystem health [46, 71]. One of the commonly observed effects of increased piscivore biomass is a reduction in planktivore biomass and an increase in herbivore biomass, as indicated by the increased dominance of Gizzard shad [19, 69]. Furthermore, it is encouraging to see that median biomass contribution has not decreased for any of the three ecotypes, pointing towards further evidence for waterfront-wide beneficial effects of restoration efforts [72–74]. Increased richness shows that habitat quality along the waterfront is improving, meeting

the needs of a wider variety of piscivores [45, 59]. These overarching positive effects are crucial when considering that large-scale restoration success depends on a variety of spatial and habitat-related aspects all interacting with each other [73].

Similar things can be said for specialist species. The high overall diversity of specialist species shows the availability of different habitat types and food sources, counteracting the general trends of increasing generalist abundances linked to habitat degradation and shifts in temperature regimes [75–78]. All three ecotypes showed a consistently high overall richness of specialist fish species (>20 species). Coastal wetlands had the lowest specialist biomass proportion, not meeting the desired 40% mark (25%, T3), while presence-absence data suggests that all specialist species, commonly expected to be found in Toronto wetlands are present [79].

These findings could be related to the fact that coastal wetlands tend to have the highest macrophyte density of the three ecotypes. Specialist species preferring cover and vegetation are normally associated with specific plant taxa. Consequently, coastal wetlands could be frequented by a higher proportion of generalist species, benefiting from the overall macrophyte presence [80]. Other factors like anthropogenic biotic homogenization (i.e., increase in community similarity through disturbances) of wetland fish communities would need to be evaluated on a site level, with coastal wetlands commonly being considered as the most degraded lost ecotype around Lake Ontario [81]. Species-level indicators point towards specific generalist species being especially abundant in coastal wetlands like Brown bullhead, increasing in median biomass contribution over time. Overall piscivore and specialist biomass contribution has increased by around 5% for both classes over time which shows that beneficial restoration effects might be masked by the higher generalist presence compared to embayments and open coast sites. Other specialist species, commonly accepted to be of low abundance in wetlands close to Toronto (e.g., Johnny darter, White bass, Longnose dace, Logperch) might not be adequality detected during sampling but given their rarity should not contribute significantly to overall biomass [79].

Finally, coastal wetlands have a wide variety of transitional species compared to resident species, with resident species tending to be more generalist (e.g., Brown bullhead). Wetlands can serve as spawning habitats for many species (e.g., Northern pike, White sucker, Walleye), as well as nursery habitats (e.g., Gizzard shad, Spottail shiner) with these species migrating in and out of the wetlands. This high temporary turnover potentially further contributes to the higher generalist biomass at this ecotype depending on sampling timeframes (e.g. summer vs fall) [79, 82].

## 4.3 Limitations and future directions

### 4.3.1 Electrofishing as primary survey method to monitor fish populations.
All commonly deployed aquatic sampling methods are associated with biases, with electrofishing commonly accepted as one of the most non-discriminant methods for assessing communities as [83, 84]. It is well known that habitat uses and behavioural differences in fish species can impact catch estimates significantly [85, 86]. Electrofishing surveys can be planned accordingly to capture spatial and temporal variation. The main goal of the monitoring program along the Toronto waterfront is to capture overall community changes in response to restoration efforts and environmental stressors hence yearly pooled single-pass surveys across ecotype replicates and reference sites yield accurate data to detect broad community or population changes [33, 84, 87]. Uncalibrated single pass surveys can yield misleading trends for single species estimates, especially for species with low detection rates [86–89]. While surveys along the waterfront do not target single species, catch efficiency bias across different species can still be a source for errors for the community [83, 85, 86, 90]. Especially swim bladder-lacking benthic

species like *Neogobius* spp. have a lower probability of being caught during electrofishing surveys, with catch efficiency differences of up to 45% compared to species with well-developed swim bladders [90]. Differences like that need to be considered when aiming to estimate total abundance of species. Alternative gears like traps and netting can help correct for that bias [83, 85]. Manual shore surveys like seining are already part of the monitoring efforts, especially at restored sites along the Toronto waterfront but usually require additional time and effort and are often not feasible in many urban riparian areas [33, 91, 92]. Alternative methods also usually cover significantly less ground compared to electrofishing surveys which plays an important role for surveying and monitoring large areas like the Greater Toronto Region [84, 86].

**4.3.2 Metrics used to assess fish communities and accounting for a changing world.** Another limitation of using large scale temporal and spatial datasets, is the need to find metrics that adequately capture and describe community and population changes [93–95]. While common metrics like CPUE and BPUE are well-established, behavioural traits and guilds are often far less tangible on a larger spatial scale [94, 96, 97]. For instance, using specialist, piscivore and generalist species and biomass thresholds as benchmarks for restoration success can be intuitive to assess waterfront wide changes, however communities and species tend to be more nuanced on an individual level [91, 96]. Many species have primary, secondary and tertiary feeding guilds. For example, Smallmouth bass are both invertivore and piscivore. Feeding guilds and other traits can also change across life history stages [95, 98–101]. As for this study, metrics and targets are derived from the official Toronto and Region Remedial Action Plan and in accordance with the Toronto Aquatic Habitat Restoration Strategy [33, 91]. The metrics were designed to capture ecological integrity and responses of native and naturalized species to restoration efforts along the shoreline and nearshore waters of the Toronto waterfront and to be accessible and of use to both the public as well as stakeholders and managers [33, 35, 74, 91, 102]. Overall, there are more and more efforts to develop holistic metrics that capture a variety of species traits and habitat requirements and preferences across various life stages that will be incredibly valuable to be implemented in future monitoring efforts, especially for individual sites [35, 37, 74, 102].

Large scale stressors like pollution, increased urbanization, imperviousness, and road density as well as Lake and watershed connectivity play an important role in the detected community changes [41, 81, 103]. While this study was aimed to capture overall community and population changes along the Toronto Waterfront, future studies will consider stressors in combination with restoration efforts to estimate their overall effect on community changes. Identifying primary drivers for community changes will be helpful to put restoration effectiveness into perspective and be able to prioritize stressors and target areas associated with said stressors [9, 68, 104, 105]. The next step will be to evaluate each restored site and its larger role within the spatial context of the waterfront under consideration of proximity to biodiversity hotspots, stressors, and interaction with adjacent sites. Furthermore, including estuary sites will further complement the assessment of the waterfront.

# 5. Conclusions and implications

## 5.1 Implications for the state of the nearshore fish community along the Toronto waterfront and associated restoration goals

Analyzing the collected long-term monitoring data for the waterfront both in terms of population trends and specific targets as well as trait-based community changes holds important implications for future management. The fish community along the Toronto waterfront has changed and is still changing towards an increased dominance of native warmwater and cool-water piscivores, associated with cover and vegetation, which is meeting fish community

targets and providing evidence that implemented restoration and management measures are effective for all three ecotypes.

Many areas show known competitive relationships–e.g., Alewife, Gizzard shad, and various shiner species that should be considered when evaluating monitoring data and this helps to understand certain trends and how population fluctuations are driven [16, 22, 58]. Common carp relative abundance and biomass have decreased at many restored sites, pointing towards the effectiveness of exclusion barriers but should be evaluated on a broader scale to estimate carp presence and biomass at more unrestored or open sites [18, 21]. While restoration efforts might not reduce the overall abundance of non-native species along the waterfront outside the restored sites, it still serves as important refuge for species in direct competition with non-native species. Restoration efforts have the potential to increase overall waterfront resilience to future invasion events and further taxonomic and food-web-related responses should become apparent in the future [17, 63, 106].

Another important consideration is the food web reliance on Alewife as indicated by many studies and reports. Lake-wide declines, picked up in our nearshore data might lead to a reduction in forage fish availability if driven to the extreme or compensated through an increase in other forage fish species [24, 25]. On the other hand, an overabundance of Alewife, initially non-native to Lake Ontario, has been associated with Lake trout declines and the previously mentioned competitive interaction with other species that could benefit from lower Alewife numbers [58, 107]. Based on our data, at nearshore sites, this seems to be the case for Gizzard shad and other smaller-bodied species like Golden shiner and Bluntnose minnow [108] but should be considered for future monitoring programs.

Still, unrestored sites along the waterfront hold immense restoration potential and already seem to benefit from neighboring restored sites in terms of community composition and population stability and can rely on previous knowledge gathered from the currently established and monitored restoration sites.

## Supporting information

**S1 Table. Catch per unit effort (CPUE) comparison across ecotypes.** Kruskal-Wallis test and conover comparison (BH adjusted p-value) for years blocked into three periods (03–08 T1, 09–14 T2, 15–21 T3).
(DOCX)

**S2 Table. Biomass per unit effort (BPUE) comparison across ecotypes.** Kruskal-Wallis test and conover comparison (BH adjusted p-value) for years blocked into three periods (03–08 T1, 09–14 T2, 15–21 T3).
(DOCX)

**S3 Table. Redundancy analysis (RDA) output (adonis, 999 permutations), comparing years blocked into three periods (03–08 T1, 09–14 T2, 15–21 T3) based on relative catch and community traits (Hellinger transformed).**
(DOCX)

**S4 Table. Redundancy analysis (RDA) output (adonis, 999 permutations), comparing years blocked into three periods (03–08 T1, 09–14 T2, 15–21 T3) based on relative biomass and community traits (Hellinger transformed).**
(DOCX)

**S5 Table. Biomass contribution to total biomass as well as richness for piscivore and specialist species comparison across ecotypes.** Kruskal-Wallis test and Dunn post-hoc (BH

adjusted p-value) for years blocked into three periods (03–08 T1, 09–14 T2, 15–21 T3).
(DOCX)

**S1 Fig. Waterfront fish species.** Waterfront fish species list and scientific names.
(TIF)

**S2 Fig. Population trend analyses.** Workflow to analyze population trends for catch (CPUE) and biomass (BPUE) over time based on summer boat electrofishing data and across ecotypes (embayment, open coast, coastal wetland) between the years of 2003 and 2021. Years were blocked into three periods (03–08, 09–14, 15–21). Symbol attribution Tracey Saxby, Integration and Application Network; Kate Moore, Moreton Bay Waterways and Catchments Partnership (ian.umces.edu/media-library). Reprinted from ian.umces.edu/media-library under a CC BY 4.0 license, with permission from.
(TIF)

**S3 Fig. Community trait and composition analyses.** Workflow to analyze community changes based on relative catch and biomass (Hellinger transformed) over time based on summer boat electrofishing data and across ecotypes (embayment, open coast, coastal wetland) between the years of 2003 and 2021. Years were blocked into three periods (03–08, 09–14, 15–21). Included traits were status, feeding guild, habitat, and thermal guild and plotted in ordination space (RDA). Symbol attribution Tracey Saxby, Integration and Application Network; Kate Moore, Moreton Bay Waterways and Catchments Partnership (ian.umces.edu/media-library). Reprinted from ian.umces.edu/media-library under a CC BY 4.0 license, with permission from.
(TIF)

**S4 Fig. Piscivore and specialist biomass analyses.** Workflow to analyze biomass contribution to total biomass for piscivore and specialist species over time based on summer boat electrofishing data and across ecotypes (embayment, open coast, coastal wetland) between the years of 2003 and 2021. Years were blocked into three periods (03–08, 09–14, 15–21). Symbol attribution Tracey Saxby, Integration and Application Network; Kate Moore, Moreton Bay Waterways and Catchments Partnership (ian.umces.edu/media-library). Reprinted from ian.umces.edu/media-library under a CC BY 4.0 license, with permission from ian.umces.edu/media-library, original copyright 2005 & 2010.
(TIF)

## Acknowledgments

We want to acknowledge the role that open-access data and scientific material play in conducting research. Monitoring data is available through TRCA, and scientific symbols used to enhance figures were provided through ian.umces.edu under the Attribution-ShareAlike 4.0 International (CC BY-SA 4.0) agreement.

## Author Contributions

**Conceptualization:** Sebastian Theis, Andrea Chreston, Angela Wallace, Brian Graham, Brynn Coey, Don Little, Lyndsay Cartwright, Mark Poesch, Rick Portiss, Jonathan Ruppert.

**Formal analysis:** Sebastian Theis, Jonathan Ruppert.

**Funding acquisition:** Mark Poesch, Jonathan Ruppert.

**Investigation:** Sebastian Theis, Lyndsay Cartwright, Jonathan Ruppert.

**Methodology:** Sebastian Theis, Andrea Chreston, Angela Wallace, Brian Graham, Brynn Coey, Don Little, Lyndsay Cartwright, Mark Poesch, Rick Portiss, Jonathan Ruppert.

**Project administration:** Sebastian Theis, Lyndsay Cartwright, Jonathan Ruppert.

**Visualization:** Sebastian Theis.

**Writing – original draft:** Sebastian Theis.

**Writing – review & editing:** Sebastian Theis, Andrea Chreston, Angela Wallace, Brian Graham, Brynn Coey, Don Little, Lyndsay Cartwright, Mark Poesch, Rick Portiss, Jonathan Ruppert.

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
