## [Decision Letter · Decision Letter 0]

2 Nov 2023

PONE-D-23-25693Nearshore fish community changes along the Toronto Waterfront in accordance with management and restoration goals:

Insights from two decades of monitoringPLOS ONE

Dear Dr. Theis,

Thank you for submitting your manuscript to PLOS ONE. After careful consideration, we feel that it has merit but does not fully meet PLOS ONE’s publication criteria as it currently stands. Therefore, we invite you to submit a revised version of the manuscript that addresses the points raised during the review process. Please submit your revised manuscript by Dec 17 2023 11:59PM. If you will need more time than this to complete your revisions, please reply to this message or contact the journal office at plosone@plos.org. Please include the following items when submitting your revised manuscript:A rebuttal letter that responds to each point raised by the academic editor and reviewer(s). You should upload this letter as a separate file labeled 'Response to Reviewers'.A marked-up copy of your manuscript that highlights changes made to the original version. You should upload this as a separate file labeled 'Revised Manuscript with Track Changes'.An unmarked version of your revised paper without tracked changes. You should upload this as a separate file labeled 'Manuscript'.

We look forward to receiving your revised manuscript.

Kind regards,

Ram Kumar, D. Sc..

Academic Editor

PLOS ONE

“Funding for this project was provided by Mitacs Cluster Accelerate. We thank the regional municipalities of Peel, York, Durham, and the City of Toronto for continued funding of waterfront fisheries monitoring and aquatic ecosystem research programs. We want to further acknowledge the role that open-access data and scientific material play in conducting research. Monitoring data is available through TRCA, and scientific symbols used to enhance figures were provided through ian.umces.edu under the Attribution-ShareAlike 4.0 International (CC BY-SA 4.0) agreement.”

“Funding for this project was provided by Mitacs Cluster Accelerate IT28524 to Dr. Mark Poesch.

4. We note that Figures 2, 3, 6, 7, 8, Fig S1, Fig S2, Fig S3 and Fig S4 in your submission contain copyrighted images. All PLOS content is published under the Creative Commons Attribution License (CC BY 4.0), which means that the manuscript, images, and Supporting Information files will be freely available online, and any third party is permitted to access, download, copy, distribute, and use these materials in any way, even commercially, with proper attribution. For more information, see our copyright guidelines: http://journals.plos.org/plosone/s/licenses-and-copyright.

1. You may seek permission from the original copyright holder of Figures 2, 3, 6, 7, 8, Fig S1, Fig S2, Fig S3 and Fig S4 to publish the content specifically under the CC BY 4.0 license.

Additional Editor Comments:

This study presents broad spatiotemporal scale monitoring data from the Toronto Waterfront nearshore community. The Toronto region has one of the highest population densities and is one of the largest urban areas in Canada affecting watershed having ∼72km of shoreline is ecologically important for aquatic biodiversity and also for Human activities along the shore. This paper determine how the fish populations along the Toronto Waterfront have changed over the past two decades. Explained Catch and biomass, and the community structure identifying most abundant and biomass-contributing species. Authors have determined if the community changes align with the most prominent stated restoration goals. by studying population trends, Community changes; and presence of piscivores and specialist species. There are some some typo errors and long sentences that need to be restructured. Some ambiguous phrases are to be rewritten or avoided.

I have some curiosities and suggestions

Curiosity: Functional feeding groups have been identified as - Herbivore, Invertevore, Planktivore and pisvivore however in natural setup Feeding guilds are not completely isolated rather one a top predator utilize food from more than one tropic levels , similar otherwise considered to omnivore or invertebrate plankton such as cyclopoid copepods may voraciously feed on Fish larvae (DOI: 10.1016/j.aquaculture.2012.01.001 ) . How to differentiate planktivory from invertivore? what about including propensity of feeding? These issue need little more elaboration;

What are specialist biomass or overall specialist richness? Is not specialist vs generalist feeding or other paradigm?

Can this paper qualify as trajectory analyses paper during restoration to monitor if restoration goals are achieved or not? Differences in community structure between restored and un restored regions can be elaborated/

regarding Objective : broad spatiotemporal scale monitoring data: Can be stated objectively: broad spatiotemporal scale can have different meaning for different context/sites

how were influential species identified?

Trend of proportional representation of alien species in restored sites may be elaborated .

Besides several sentences are too long such as lines : 63-67 Page3: more than one subordinate clause in sentences may be restructured .

Lines 102-104: Can be shifted to Method section with separate sun heading of Study site . It be better structured if the location map of study site and other details sabout field could be given under this heading in Method section.

Conclusion:

These results can be beneficial to both managers, decisions makers and other stake holders involved in freshwater restoration project to better inform and evaluate if the nearshore community along the Toronto Waterfront is approaching the outlined restoration targets and desired community composition and realization of the importance of high-quality datasets collection for multi-decade timeframes.

These results may be explained in for global freshwater restoration project; as stated this study focuses on the importance of large-scale freshwater restoration efforts within the Great Lakes. So for application it will not be confined to Lake Ontario, as the most downstream lake, rather restoration of freshwater great lakes after negative impacts exerted through landuse, development projects and overarching processes like climate change and globalization may be mitigated in other lakes and monitored using the approach herein.

Reviewers' comments:

Reviewer's Responses to Questions

**Comments to the Author**

1. Is the manuscript technically sound, and do the data support the conclusions?

Reviewer #1: Yes

Reviewer #2: Yes

2. Has the statistical analysis been performed appropriately and rigorously? 

Reviewer #1: Yes

Reviewer #2: Yes

3. Have the authors made all data underlying the findings in their manuscript fully available?

Reviewer #1: Yes

Reviewer #2: Yes

4. Is the manuscript presented in an intelligible fashion and written in standard English?

Reviewer #1: No

Reviewer #2: Yes

5. Review Comments to the Author

Reviewer #1: The article has well evaluated the spatial-temporal data of the restoration programs contributing to the effective conservation of shoreline and riparian habitat. However, the article contains large sentences and grammatical errors making some lines difficult to understand. Please double check the entire manuscript.

Line No:82 Only genus and species names are written in italics.

Line No:83-84 it is difficult to understand “While these and other impacts and events are considered lake-wide, they can manifest differently in a pelagic or littoral context” Authors should avoid ambiguous words and consider rewriting the sentence.

Line No: 87 it should be “nutrient” in place of “energy”.

Line No: 88-92 I don’t mean what authors want to say using “said changes” also the sentence is too long and better to rephase.

Line No: 94-98 “A decline of specific species in one area, for example, nearshore Alewife occurrence, can be driven by impacts in another area, the lake-wide offshore decline in Alewife, and might make restoration and enhancement efforts at the specific site less effective or vice versa mask the actual benefits of a restored site when focusing on species or metrics driven by outside factors”. This sentence is too long and better to rephase.

Line No: Authors should write exact number of restoration sites instead of “dozens of restoration sites”

Line No:167 The abbreviations used in the article for the first time should be clearly defined. It should be “Biomass per unit effort” instead of “biomass per run”.

Line no:162-166 Sentence is not clear; authors should consider rephrasing the sentence.

Line no: 206 Is there any other time intervals apart from T1, T2 and T3?

Line No: 237 Author should remove “and”.

Line NO: 265 “especially comparing T1 and T3 and T2 and T3” I don’t understand what authors wants to say?

Line No: 268 Author should remove “and” and it should be “BPUE” instead of “CPUE”

Line 278: Sentence is not clear; authors should consider rephrasing the sentence.

Line NO: 333 Authors should use T1, T2 and T3 in place of time periods “03-08 yellow, 09-14 green, 15-21 blue”.

Line No: 369 Something is missing after “19%”.

Line No: Sorry I don’t understand by “strong Alewife”, what does authors want to say?

Line No:467-470 & 472-477 These sentences are too long. Please rephrase.

Line No: 636 Authors should mention date (Assessed on) when the data/report were assessed from any website.

I would suggest to write T1, T2 and T3 instead of past, intermediate and present in fig 2,3,6 and 7 to have better understanding of graphs.

In fig-3, axis titles should be in accordance to figure legend. Further, resolution of graphs can be enhanced.

Reviewer #2: • The study design is well-thought-out and the methods are sound. The authors have clearly addressed the limitations of previous studies and have developed a new approach that is more efficient and humane.

• The results are clear and convincing.

• The paper is well-written and easy to follow. The authors have done a good job of explaining the methods and results in detail.

6. PLOS authors have the option to publish the peer review history of their article (what does this mean?). If published, this will include your full peer review and any attached files.

Reviewer #1: **Yes: **Devesh Kumar Yadav

Reviewer #2: No

---

## [Author Response · Author response to Decision Letter 0]

12 Dec 2023

We would like to thank Reviewers as well as the Editor for taking the time and effort necessary to review the manuscript both in terms of content and form. We sincerely appreciate all valuable comments and suggestions, which helped us to improve the quality and accessibility of the manuscript.

Comments:

1. Overall, the manuscript "Nearshore fish community changes along the Toronto Waterfront in accordance with management and restoration goals: Insights from two decades of monitoring" is a well-written and informative study. 

2. The research aimed at broad spatiotemporal scale monitoring data from the Toronto Waterfront nearshore community to determine how the fish populations along the Toronto Waterfront have changed over the past two decades.

3. The authors use a long-term dataset of electrofishing data to examine changes in the nearshore fish community of the Toronto Waterfront. They identify a few key species that are driving changes in the community, and they discussed the potential implications of their findings for future management and restoration efforts.

4. The research data supports the conclusions and the experiments have been conducted with appropriate methodology and proper sample sizes based on appropriate data available.

5. The statistical analysis has been performed appropriately and rigorously.

6. The study focused on fish communities along the waterfront based on relative catch and biomass have changed towards an increased dominance of native warmwater species, preferring vegetation and cover as well as an increase in piscivore presence.

7. This study will be beneficial to biologists, policy makers and other stakeholders in understanding key relationships of species and associated traits that are related to biodiversity and fish community changes. This will also help in restoration of the home of native fish species and provide them suitable ecosystem.

Review Comments to the Author

Feedback/ Suggestions:

1. The manuscript is well structured and data analysis is done with proper data collection.

2. In results (Line no. 248, 249, 279-287, 291), uniformity may be maintained in writing time periods such as T1-T2, T1-T3, T2-T3 instead writing T2-T1, T3-T1, T3-T2 etc.

Response:

Thank you kindly, that is an excellent suggestion. We have adjusted the time periods to ensure uniformity. Please refer to lines 246-247 within the tracked changes document.

3. In results (Line no. 283), biomass declined over time (T2-T1 p < n.s. should be corrected with T1-T2 p = n.s. 

Response:

 Has been changed. Please refer to lines 278-286 and 290 within the tracked changes document.

4. In Discussion (Line no. - 377), increase in herbivores (+7% median biomass contribution, Fig 8). But, Fig. 8 depicts +9% median biomass contributions of herbivores. Similarly (Line no. - 402), a decrease in planktivores (-5% median biomass contribution, Fig 8). But, Fig. 8 depicts -14% median biomass contributions of planktivores. Just check it and make necessary corrections if required. 

Response:

 Thanks for catching that, that was a mix-up with the generalist species which are 7%, has been corrected. Please refer to lines 376 and 401/402 within the tracked changes document.

5. In Discussion (Line no. 429-233), Sentence “Wide variation in Alewife cohorts …………… in nutrient input as well as anthropogenic development” may be rewritten with proper clause and syntax for better understandings of authors’ view. It may be rephrased into two sentences.

Response:

Has been edited/ rephrased into two sentences. Please refer to lines 429-433 within the tracked changes document.

6. In Discussion (Line no. 467-470), Sentence “Embayments field a broader mix of warmwater and coolwater species ………. exposed to upwellings?” may be rewritten with proper clause and syntax for better understandings of authors’ view. It may be rephrased into two sentences.

Response: 

We fully agree to split the sentence to increase coherence and comprehension. Please refer to lines 467-471 within the tracked changes document.

7. In references, line no. 613, name of a co-author should be corrected as Lapointe NW;

Response: 

Has been corrected.

8. In the same reference (line no. 614), Cyprinus carpio should be written in italics.

Response: 

Has been corrected.

9. There is need to discuss the limitations of the study in more detail, such as the potential bias of electrofishing and the lack of data on other factors that could be driving changes in the fish community.

Response: 

Thank you for this insightful suggestion. We agree that adding more information and consideration for limitations is beneficial to the broader understanding of the study. We have added limitations with a special focus on e-fishing as the primary sampling method to the manuscript. Please refer to 4.3 in the tracked changes document. There is also ongoing research, as this study is the first basis for multiple follow up studies, that uses precision and power analysis to further evaluate the current sampling design as well as the ability to detect specific species which will hopefully benefit managers in the future.

10. The authors discuss the potential implications of their findings for future management and restoration efforts. For example, they could discuss which species or ecotypes may be most vulnerable to future changes and how these species or ecotypes could be prioritized in restoration efforts.

Response: 

Thank you for this interesting thought as it overlaps very much with ongoing and future research in the region. We agree that adding vulnerable species or ecotypes to the discussion would be beneficial. Species-specific monitoring and management are being done in the region e.g., for Redside dace on a more local level under the species at risk act (SARA). In that sense these are two different legislative and jurisdictional cases whereas the used data is based on the RAP and delisting process of Toronto and Region as an overall impaired area. In terms of ecotypes it is commonly accepted that coastal wetlands face the most rapid loss within the region while normally containing the highest biodiversity as well as providing essential spawning and rearing habitat. Follow up research is looking at site specific community changes to capture these impacts as well as the development of more holistic community metrics that capture a more complete set of fish behavior, life-history and habitat preferences and requirements. 

11. The study does not account for all of the potential factors that could be driving changes in the fish community, such as climate change and pollution. It is important to keep these other factors in mind when interpreting the results of the study.

Response: 

We fully agree to this and have added a few discussion points to the previously suggested limitations section as we feel that not accounted factors like pollution and climate change fit there the best. Future studies linked to this monitoring program will consider broader spatial components like urbanization or watershed connectivity and could account more for climate variables as well as pollution or other factors not considered here. Please refer to lines 533 to 582 of the tracked changes document.

At last, I request the authors to kindly make the necessary corrections before final submission of the manuscript.

Reviewer Recommendation:

1. I recommend the manuscript for publication in PLOS ONE, with minor revisions.

2. I appreciate the authors' efforts to account for different ecotypes and to identify the key species driving changes in the community.

3. The manuscript presents a novel approach to fish habitat restoration and aquatic biodiversity conservation that is more precise, efficient, and humane. 

4. The paper is well-written and easy to follow. The study is clear and convincing. 

5. Overall, the manuscript is well-written and informative. I recommend that the authors may revise the manuscript to address the comments and to include the additional suggestions that I have made.

Review Comments to the Author

Reviewer #1: The article has well evaluated the spatial-temporal data of the restoration programs contributing to the effective conservation of shoreline and riparian habitat. However, the article contains large sentences and grammatical errors making some lines difficult to understand. Please double check the entire manuscript.

Line No:82 Only genus and species names are written in italics.

Response:

Has been corrected, thanks. Please refer to lines 79-80 within the tracked changes document.

Line No:83-84 it is difficult to understand “While these and other impacts and events are considered lake-wide, they can manifest differently in a pelagic or littoral context” Authors should avoid ambiguous words and consider rewriting the sentence.

Response:

Thank you for the suggestion. We have edited the sentence accordingly. Please refer to lines 81-83 within the tracked changes document.

Line No: 87 it should be “nutrient” in place of “energy”.

Response:

Has been replaced. Please refer to lines 81-83 within the tracked changes document.

Line No: 88-92 I don’t mean what authors want to say using “said changes” also the sentence is too long and better to rephase.

Response:

Thanks for pointing that out. ‘Said changes’ refers to the previously described changes in nutrient loads between nearshore and offshore habitats. We have edited the sentence to clarify that and split the sentence in two. Please refer to lines 87-91 within the tracked changes document.

Line No: 94-98 “A decline of specific species in one area, for example, nearshore Alewife occurrence, can be driven by impacts in another area, the lake-wide offshore decline in Alewife, and might make restoration and enhancement efforts at the specific site less effective or vice versa mask the actual benefits of a restored site when focusing on species or metrics driven by outside factors”. This sentence is too long and better to rephase.

Response:

Has been rephrased, thanks. Please refer to lines 93-97 within the tracked changes document.

Line No: Authors should write exact number of restoration sites instead of “dozens of restoration sites”

Response:

Has been added (44 restoration projects), thanks. Please refer to lines 152 & 159 within the tracked changes document.

Line No:167 The abbreviations used in the article for the first time should be clearly defined. It should be “Biomass per unit effort” instead of “biomass per run”.

Response:

Thanks for catching that, has been corrected. Please refer to lines 167-168 within the tracked changes document.

Line no:162-166 Sentence is not clear; authors should consider rephrasing the sentence.

Response:

These sentences refer to the way that abundant species are sampled. For instance, if 50 Yellow perch are caught in the same run only the first 20 are weighed and measured individually. The remaining 30 are weighed as one batch altogether and the largest and smallest individual length is taken. Overall, this information might not really add anything in terms of essential information. Thus, we have restructured and shortened the sentence. Please refer to lines 162-168 within the tracked changes document.

Line no: 206 Is there any other time intervals apart from T1, T2 and T3?

Response:

There is data for monitoring between 1989 and 2002 but sampling during that time was sporadic and not standardized hence cannot be used for accurate catch or biomass trends.

Line No: 237 Author should remove “and”.

Response:

Has been removed, thanks. Please refer to line 237 within the tracked changes document.

Line NO: 265 “especially comparing T1 and T3 and T2 and T3” I don’t understand what authors wants to say?

Response:

Thanks for pointing that out. The ‘especially’ just emphasizes that the largest declines in biomass contribution from Carp and Freshwater drum occurred when comparing T1 and T2 to T3. Has been removed to avoid confusion. Please refer to line 265 within the tracked changes document.

Line No: 268 Author should remove “and” and it should be “BPUE” instead of “CPUE”

Response:

Has been corrected, thanks. Please refer to line 268 within the tracked changes document.

Line 278: Sentence is not clear; authors should consider rephrasing the sentence.

Response:

Has been edited. Please refer to line 278-284 within the tracked changes document.

Line NO: 333 Authors should use T1, T2 and T3 in place of time periods “03-08 yellow, 09-14 green, 15-21 blue”.

Response:

We fully agree. T1, T2 and T3 have been added aside from the color distinction. 

Added T1-T3 to all figures.

Line No: 369 Something is missing after “19%”.

Response:

Has been corrected – ‘T3’ moved from the brackets to the ‘19%’. Please refer to line 370 within the tracked changes document.

Line No: Sorry I don’t understand by “strong Alewife”, what does authors want to say?

Response:

It refers to the yearly cohort strength in terms of reproductive output from one year to the next, e.g., a cohort refers to fish with a similar age. So, a strong reproductive year for instance for Alewife means that there will be a very ‘strong’ age 1 cohort the next year as those young fish grow older and thus more competition for Emerald shiner. Please refer to line 437 within the tracked changes document.

Line No:467-470 & 472-477 These sentences are too long. Please rephrase.

Response:

Has been broken up, not multiple sentences. Please refer to line 467-477 within the tracked changes document.

Line No: 636 Authors should mention date (Assessed on) when the data/report were assessed from any website.

Response:

Date has been added as ‘last assessed on’. 

I would suggest to write T1, T2 and T3 instead of past, intermediate and present in fig 2,3,6 and 7 to have better understanding of graphs.

Response:

Thanks, excellent suggestion. Has been adjusted for all figures.

In fig-3, axis titles should be in accordance to figure legend. Further, resolution of graphs can be enhanced.

Response:

Thanks for this suggestion. We have edited the figure legend accordingly. Original figures are all in 600 dpi, but we acknowledge that the journal portal created pdf reduces the resolution significantly. 

Reviewer #2: • The study design is well-thought-out and the methods are sound. The authors have clearly addressed the limitations of previous studies and have developed a new approach that is more efficient and humane.

• The results are clear and convincing.

• The paper is well-written and easy to follow. The authors have done a good job of explaining the methods and results in detail.

Additional Editor Comments:

This study presents broad spatiotemporal scale monitoring data from the Toronto Waterfront nearshore community. The Toronto region has one of the highest population densities and is one of the largest urban areas in Canada affecting watershed having ∼72km of shoreline is ecologically important for aquatic biodiversity and also for Human activities along the shore. This paper determine how the fish populations along the Toronto Waterfront have changed over the past two decades. Explained Catch and biomass, and the community structure identifying most abundant and biomass-contributing species. Authors have determined if the community changes align with the most prominent stated restoration goals. by studying population trends, Community changes; and presence of piscivores and specialist species. There are some some typo errors and long sentences that need to be restructured. Some ambiguous phrases are to be rewritten or avoided.

Response:

Thanks for this suggestion. As mentioned by the reviewers we have edited and revised certain parts and sentences to correct typos, errors and increase accessibility and readability throughout the text. 

I have some curiosities and suggestions

Curiosity: Functional feeding groups have been identified as - Herbivore, Invertevore, Planktivore and pisvivore however in natural setup Feeding guilds are not completely isolated rather one a top predator utilize food from more than one tropic levels , similar otherwise considered to omnivore or invertebrate plankton such as cyclopoid copepods may voraciously feed on Fish larvae (DOI: 10.1016/j.aquaculture.2012.01.001 ) . How to differentiate planktivory from invertivore? what about including propensity of feeding? 

Response:

We fully agree on this. We have added a limitation and future directions section to the manuscript. Feeding guild refers to primary feeding guild while fully acknowledging that most species also have secondary and tertiary feeding guilds which can also change across their respective lie history. We have discussed this shortcoming in the mentioned limitations section as well as elaborated that future and ongoing studies which focus more on individual sites are working towards a better incorporation of more holistic metrics (e.g., life history traits vs habitat matrices). This study is the first step to capture broad community changes in accordance with overall restoration goals and includes 3 ongoing follow up studies targeting 1) Sampling accuracy and precision 2) Spatial interactions within a watershed and landscape network (e.g., urbanization, road density) 3) Developing better metrics and indices to capture community changes on a more holistic level.

These issue need little more elaboration;

What are specialist biomass or overall specialist richness? Is not specialist vs generalist feeding or other paradigm?

Response:

That is correct. So given the large urban area that TRCA and other agencies are working in they wanted to develop intuitive benchmarks thus assigning each fish species to one of three groups (piscivore, specialist, generalist). Piscivore predominantly feed on other fish, specialists focus primarily on one feeding guild EXCEPT piscivores (invertivore, planktivory, etc.) and generalists are opportunistic. So, each fish species that can be found along the waterfront has an assigned label. The overall goal here by NGOs and stakeholders was to use specialists and piscivores as a proxy for a healthy food web and trophic stability. Aiming for 20% piscivores and 40% specialist ensures a balanced food web and should respond to restoration efforts e.g., providing diverse habitat through large woody debris, shoreline revetting, creation of coastal wetlands and riparian planting etc.

The 20% and 40% refers to biomass contribution in each area by those groups. However, it is also of interest to assess richness as e.g., theoretically 20% piscivore biomass contribution could be all due to a single species which then goes against the overall goal to establish a diverse and balanced food web. Overall, those are the reasons to look both at biomass and richness.

Ideally as mentioned before, future metrics that go beyond the current restoration plan implemented by the government will include primary, secondary, and tertiary feeding guilds as well as more habitat and life-history traits which is currently being developed.

In combination with reviewer comments we have edited some of the methods/ method section as well as shortened or expanded certain sampling and data analyses steps as outlined in the reviewer replies.

Can this paper qualify as trajectory analyses paper during restoration to monitor if restoration goals are achieved or not? Differences in community structure between restored and un restored regions can be elaborated/

regarding Objective : broad spatiotemporal scale monitoring data: Can be stated objectively: broad spatiotemporal scale can have different meaning for different context/sites

how were influential species identified?

Trend of proportional representation of alien species in restored sites may be elaborated .

Response:

Absolutely. Results based on biomass thresholds and community composition are being used to decide whether the waterfront can be delisted as a BUI (beneficial use impairment). Future studies focus more on the individual sites and restoration projects like Gibraltar point, TTP and Frenchman’s Bay. So, you are right, these approaches and results will be applied not only for the waterfront as a whole but also to evaluate site specific restoration success and community changes. These site-specific assessments will feature more in-depth metrics building mostly on previous work from the department of fisheries and oceans e.g., Minns et al. and their effort to create life-history habitat matrices. Site-specific assessment furthermore will consider specific presence of non-native species e.g., efforts to keep Common Carp out of wetlands through exclusion barriers or vice versa the specific provision of spawning habitat (e.g., spawning channels for Northern Pike) all the way to removal of invasive riparian plants and macrophytes. Site-specific efforts are supported by high resolution methods like telemetry tracking data. There are distinct changes in benthic macroinvertebrate communities as well as restored sites which we are investigating in different studies. So overall we fully agree with all your points and are looking forward to covering those in the mentioned follow up studies which are underway now.

Besides several sentences are too long such as lines : 63-67 Page3: more than one subordinate clause in sentences may be restructured .

Response:

Sentence structure and length has been revised for many sections of the text.

Lines 102-104: Can be shifted to Method section with separate sun heading of Study site . It be better structured if the location map of study site and other details sabout field could be given under this heading in Method section.

Response:

Excellent suggestions, text and captions have been moved.

Conclusion:

These results can be beneficial to both managers, decisions makers and other stake holders involved in freshwater restoration project to better inform and evaluate if the nearshore community along the Toronto Waterfront is approaching the outlined restoration targets and desired community composition and realization of the importance of high-quality datasets collection for multi-decade timeframes.

Response:

We couldn’t agree more. We are conducting a precision and accuracy analysis right now to further determine how much sampling is needed to capture community changes properly to be able to have high quality data. Data has been collected since the 90s but was not standardized enough in terms of spatial and temporal extent, so we strive towards consistently improving the monitoring program.

These results may be explained in for global freshwater restoration project; as stated this study focuses on the importance of large-scale freshwater restoration efforts within the Great Lakes. So for application it will not be confined to Lake Ontario, as the most downstream lake, rather restoration of freshwater great lakes after negative impacts exerted through landuse, development projects and overarching processes like climate change and globalization may be mitigated in other lakes and monitored using the approach herein.

Response:

Ideally results from this study and the three other ones will help us to create a comprehensive evaluation of the restoration effects over time but also to help improve the monitoring framework which could be applied to other areas. We have expanded on these points a bit more in our limitations and future direction section.

We want to thank the reviewers and editor their support once again and for this incredibly helpful discussion and overall process.

---

## [Decision Letter · Decision Letter 1]

23 Jan 2024

Nearshore fish community changes along the Toronto Waterfront in accordance with management and restoration goals: Insights from two decades of monitoring

PONE-D-23-25693R1

Dear Dr. Sebastian Theis

We’re pleased to inform you that your manuscript has been judged scientifically suitable for publication and will be formally accepted for publication once it meets all outstanding technical requirements.

Kind regards,

Ram Kumar, Ph.D.

Academic Editor

PLOS ONE

Reviewers' comments:

Reviewer's Responses to Questions

**Comments to the Author**

1. If the authors have adequately addressed your comments raised in a previous round of review and you feel that this manuscript is now acceptable for publication, you may indicate that here to bypass the “Comments to the Author” section, enter your conflict of interest statement in the “Confidential to Editor” section, and submit your "Accept" recommendation.

Reviewer #1: All comments have been addressed

Reviewer #2: (No Response)

2. Is the manuscript technically sound, and do the data support the conclusions?

Reviewer #1: Yes

Reviewer #2: Yes

3. Has the statistical analysis been performed appropriately and rigorously? 

Reviewer #1: Yes

Reviewer #2: Yes

4. Have the authors made all data underlying the findings in their manuscript fully available?

Reviewer #1: Yes

Reviewer #2: Yes

5. Is the manuscript presented in an intelligible fashion and written in standard English?

Reviewer #1: Yes

Reviewer #2: Yes

6. Review Comments to the Author

Reviewer #1: (No Response)

Reviewer #2: A. Kindly review the comment no. 8 in the reference (reference No. 18 & line no. 614), write Cyprinus carpio in italics. (Reviewer 1).

B. Replies of the remaining comments are justifiable.

C. I recommend the manuscript for publication in PLOS ONE.

7. PLOS authors have the option to publish the peer review history of their article (what does this mean?). If published, this will include your full peer review and any attached files.

Reviewer #1: **Yes: **Devesh Kumar Yadav

Reviewer #2: **Yes: **Dr. Satendra Kumar

---

## [Editor Report · Acceptance letter]

15 Feb 2024

PONE-D-23-25693R1 

PLOS ONE

Dear Dr. Theis, 

I'm pleased to inform you that your manuscript has been deemed suitable for publication in PLOS ONE. Congratulations! Your manuscript is now being handed over to our production team.

Kind regards, 

on behalf of

Professor Ram Kumar 

Academic Editor

PLOS ONE